# Hide & Seek: Transformer Symmetries Obscure Sharpness & Riemannian Geometry Finds It

**Marvin F. da Silva** [1,2]   **Felix Dangel** [2]   **Sageev Oore** [1,2]

## Abstract

The concept of sharpness has been successfully applied to traditional architectures like MLPs and CNNs to predict their generalization. For transformers, however, recent work reported weak correlation between flatness and generalization. We argue that existing sharpness measures fail for transformers, because they have much richer symmetries in their attention mechanism that induce directions in parameter space along which the network or its loss remain identical. We posit that sharpness must account fully for these symmetries, and thus we redefine it on a quotient manifold that results from quotienting out the transformer symmetries, thereby removing their ambiguities. Leveraging tools from Riemannian geometry, we propose a fully general notion of sharpness in terms of a geodesic ball on the symmetry-corrected quotient manifold. In practice, we need to approximate the geodesics. Doing so up to first order yields existing adaptive sharpness measures, and we demonstrate that including higher-order terms is crucial to recover correlation with generalization. We present results on diagonal nets with synthetic data and show that our geodesic sharpness reveals strong correlation with generalization for real-world transformers on both text and image classification tasks.

## 1. Introduction

Predicting generalization of neural nets (NNs)—the discrepancy between training and test set performance—remains an open challenge. Generalization-predictive metrics are valuable though: they enable explicit regularization of training to enhance generalization (Foret et al., 2021), and provide broader theoretical insights into generalization itself.

There is a long history of hypotheses linking sharpness to generalization, but evidence has been conflicting (Hochreiter & Schmidhuber, 1994; Andriushchenko et al., 2023). Generalization has been speculated as correlating with flatness, but recent evidence has indicated that, in the case of transformers, it has little to no correlation whatsoever. Measures of sharpness have varied widely, ranging from trace of the Hessian to worst-case loss within a local neighborhood, with adaptive and relative variations proposed to address specific challenges (Kwon et al., 2021; Petzka et al., 2021).

We suspect that some of the confusion stems from the specificity of the problem these measures have attempted to address: the issue of parameter rescaling. In contrast, we argue that rescaling (Dinh et al., 2017) is merely a special case of a broader, more fundamental obstacle to measuring sharpness accurately: the presence of full and continuous parameter symmetries. Addressing this challenge is crucial to ensure that we are studying the right quantity when investigating the relationship between sharpness and generalization.

Beyond discrete permutation symmetries, neural nets naturally exhibit continuous symmetries in their parameter space. These symmetries are intrinsic, data-independent properties that emerge from standard architectural components. For example: normalization layers (Ioffe & Szegedy, 2015; Ba et al., 2016; Wu & He, 2018) induce scale invariance on the pre-normalization weights (Salimans & Kingma, 2016); homogeneous activation functions like $\mathrm{ReLU}$ introduce re-scaling symmetries between pre- and post-activation weights (Dinh et al., 2017); some normalization layers and softmax impose translation symmetries in the preceding layer's biases (Kunin et al., 2021). As a result, arguably almost any NN, along with its corresponding loss, exhibit symmetries and can therefore represent the *same* function using *different* parameter values (Figure 1a).

Adaptive flatness (Kwon et al., 2021) accounts for some symmetries, both element- and filter-wise re-scaling, but fails to capture the attention mechanism's *full* symmetry, represented by $\mathrm{GL}(h)$ (re-scaling by invertible $h \times h$ matrices, where $h$ is the hidden dimension), as we will discuss later. Aiming to break the cycle between discovery of a

[1]Faculty of Computer Science, Dalhousie University, Halifax, Canada [2]Vector Insitute for Artificial Intelligence, Toronto, Canada. Correspondence to: Marvin F. da Silva <marvinf.silva@dal.ca>.

*Proceedings of the 42$^{nd}$ International Conference on Machine Learning*, Vancouver, Canada. PMLR 267, 2025. Copyright 2025 by the author(s).

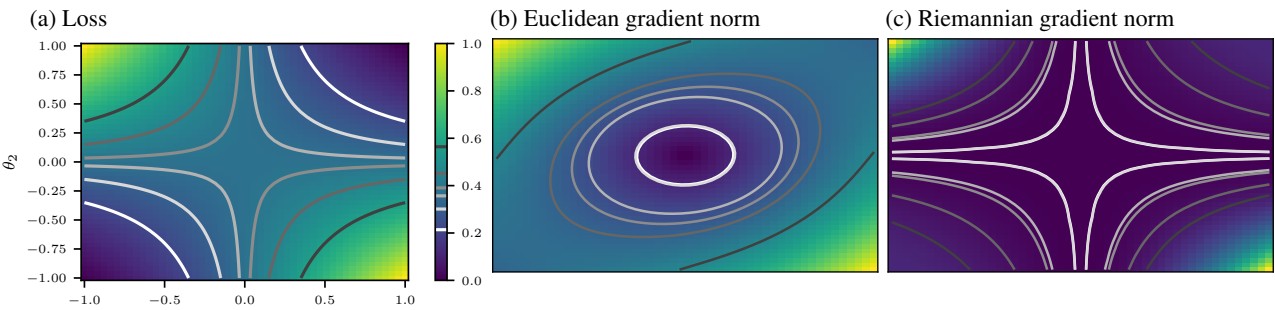

Figure 1: **Quantities from the Riemannian quotient manifold respect the loss landscape's symmetry; Euclidean quantities do not.** We illustrate this here for a synthetic least squares regression task with a two-layer NN, where $x \mapsto \theta_2 \theta_1 x$ with scalar parameters $\boldsymbol{\theta} \in \mathbb{R}^2$ and input $x \in \mathbb{R}$ (i.e. each layer is a linear function). The NN is re-scale invariant, i.e. has GL(1) symmetry: For any $\alpha \in \mathbb{R} \setminus \{0\}$, the parameters $(\theta_1', \theta_2') = (\alpha^{-1}\theta_1, \alpha\theta_2)$ represent the same function. (a) The loss function inherits this symmetry and has hyperbolic level sets. (b) The Euclidean gradient norm does not share the loss function's geometry and changes throughout an orbit where the NN function remains constant. (c) The Riemannian gradient norm follows the loss function's symmetry and remains constant throughout an orbit, i.e., it does not suffer from ambiguities for two points in parameter space that represent the same NN function.

specific symmetry and techniques to deal with it, we ask:

*Can we provide a one-size-fits-many recipe for developing symmetry-invariant quantities for a wider range of continuous symmetries?*

Here, we positively answer this question by proposing a principled approach to eliminate ambiguities stemming from symmetry. Essentially, this boils down to using the geometry that correctly captures symmetry-imposed parameter equivalences. We apply concepts from Riemannian geometry to work on the Riemannian quotient manifold implied by a symmetry group (Boumal, 2023, §9). We thus identify objects on the quotient manifold—like the Riemannian metric and gradient—and show how to translate them back to the Euclidean space. Our contributions are the following:

(a) We introduce the application of Riemannian geometry (Boumal, 2023) to the study of NN parameter space symmetries by using geometry from the quotient manifold induced by a symmetry as a general recipe to remove symmetry-induced ambiguities in parameter space. We do so by translating concepts like gradients from the quotient manifold back to the original space through *horizontal lifts*.

(b) Based on (a), we propose and analyze *geodesic sharpness*, a novel adaptive sharpness measure: By Taylor-expanding our refined geometry, we show that (i) symmetries introduce curvature into the parameter space, which (ii) results in previous adaptive sharpness measures when ignored. Geodesic sharpness differs from traditional sharpness measures in two key aspects: (i) the norm of the perturbation parameter is redefined to reflect the underlying geometry; (ii) perturbations

follow geodesic paths in the quotient manifold rather than straight lines in the ambient space.

(c) For diagonal nets, we analytically solve *geodesic sharpness* and find a strong correlation with generalization. Then, we apply our approach to the unstudied and higher-dimensional GL($h$) symmetry in the attention mechanism. On both large vision transformers and language models, we empirically find stronger correlation than any previously seen (that we are aware of) between our geodesic sharpness and generalization.

## 2. Related Work

**Symmetry versus reparameterization:** Kristiadi et al. (2023) pointed out how to fix ambiguities stemming from reparameterization, i.e. a change of variables to a *new* parameter space: Invariance under reparameterization follows by correctly transforming the (often implicitly treated) Riemannian metric into the new coordinates. Our work focuses on invariance of the parameter space $\overline{\mathcal{M}}$ under a symmetry group $\mathcal{G}$ with action $\boldsymbol{\psi} : \mathcal{G} \times \overline{\mathcal{M}} \to \overline{\mathcal{M}}$, $(g, \boldsymbol{\theta}) \mapsto \boldsymbol{\psi}(g, \boldsymbol{\theta})$ that operates on a *single* parameter space.

**Symmetry teleportation:** Another ways to use symmetry-implied ambiguity is to view it as a degree of freedom and develop adaptation heuristics to improve algorithms which are not symmetry-agnostic (Zhao et al., 2022a).

**Geometric constraints & NN dynamics:** Previous studies analyze how parameter space symmetries impose geometric constraints on derivatives and introduce conserved quantities during training (Kunin et al., 2021). Our approach

differs by systematically removing symmetry-induced ambiguity through quotienting out the the symmetry group.

We generalize earlier post-hoc solutions for simpler symmetries (e.g., $\mathrm{GL}(1)$) to more complex, higher-dimensional symmetries such as $\mathrm{GL}(h)$, common in neural network attention mechanisms. Unlike Kunin et al. (2021), who consider geometry in augmented spaces for simpler symmetries, we directly use the quotient space geometry. Objects are then 'lifted' back into the original space, yielding symmetry-corrected quantities. This method provides a principled framework capable of handling high-dimensional symmetries, leading to a more effective dimensionality reduction.

**Quotient manifolds in deep neural networks:** Rangamani et al. (2019) introduce a quotient manifold construction for re-scaling symmetries and then use the Riemannian spectral norm as a measure of worst-case flatness. This differs from our approach in several ways:

(a) Our approach is more general and contains both the $\mathrm{GL}(h)$ symmetry of transformers, and the original re-scaling/scaling symmetry of CNNs/MLPs, rendering it applicable to a wider range of modern architectures.

(b) Our experimental setup is more challenging: we test on large-scale models (large transformers vs CNNs) and large-scale datasets (ImageNet vs CIFAR-10). Sharpness measures that account for re-scaling/scaling symmetries (e.g. adaptive sharpness) work quite well on CIFAR-10 with CNNs, and tends to break down on datasets like ImageNet with transformers.

(c) Conceptually, Rangamani et al. (2019) defines worst-case sharpness on the usual norm-ball, appropriately generalized to the Riemannian setting. We propose instead that the ball should be the one traced out by geodesics, to better respect the underlying geometry.

(d) Performance-wise, our approach is cheaper as it does not use the Hessian and only uses symmetry-corrected gradients (see Dagréou et al. (2024) for an in-depth cost comparison of computing Hessians vs gradients).

**Relative sharpness:** Another promising approach to sharpness was proposed by Petzka et al. (2021), where the generalization gap is shown to admit a decomposition into a representativeness term and a feature robustness term. Focusing on the feature robustness term, they introduce relative sharpness, which is invariant to a layer- and neuron-wise re-scaling, and performs better than traditional sharpness measures (Adilova et al., 2023; Walter et al., 2025).

## 3. Preliminary Definitions, Notation & Math

**Generalization measures:** We consider a neural net $f_{\boldsymbol{\theta}}$ with parameters $\boldsymbol{\theta} \in \mathbb{R}^d$ that is trained on a data set $\mathbb{D}_{\text{train}}$ using a loss function $\ell$ by minimizing the empirical risk

$$L_{\mathbb{D}_{\text{train}}}(\boldsymbol{\theta}) \coloneqq \frac{1}{|\mathbb{D}_{\text{train}}|} \sum_{(\boldsymbol{x},\boldsymbol{y}) \in \mathbb{D}_{\text{train}}} \ell(f_{\boldsymbol{\theta}}(\boldsymbol{x}), \boldsymbol{y}) \,.$$

Our goal is to compute a quantity on the training data that is predictive of the network's generalization, i.e. performance on a held-out data set.

**Sharpness:** A popular way to predict generalization is via sharpness—i.e., how much the loss changes when perturbing the weights in a small neighbourhood—like average ($S_{\text{avg}}$) or worst-case sharpness ($S_{\text{max}}$) (Keskar et al., 2017)

$$S_{\text{avg}} = \mathbb{E}_{\mathbb{S}}\left[L_{\mathbb{S}}(\boldsymbol{\theta} + \boldsymbol{\delta}) - L_{\mathbb{S}}(\boldsymbol{\theta})\right], \quad \boldsymbol{\delta} \sim \mathcal{N}(\mathbf{0}, \rho^2 \boldsymbol{I})\,,$$

$$S_{\text{max}} = \mathbb{E}_{\mathbb{S}}\left[\max_{\|\boldsymbol{\delta}\|_2 \leq \rho} (L_{\mathbb{S}}(\boldsymbol{\theta} + \boldsymbol{\delta}) - L_{\mathbb{S}}(\boldsymbol{\theta}))\right]\,,$$

with batches $\mathbb{S} \sim \mathbb{D}_{\text{train}}$ of size $|\mathbb{S}| = m$, neighbourhood size $\rho$, and perturbation $\boldsymbol{\delta}$. Near critical points, they closely relate to the Hessian $\boldsymbol{H}$ (and thus parameter space curvature): $S_{\text{avg}} \propto \text{Tr}(\boldsymbol{H})$, and $S_{\text{max}} \propto \lambda_{\text{max}}(\boldsymbol{H})$.

**Adaptive sharpness:** Hessian-based sharpness measures can be made to assume arbitrary values by rescaling parameters, even though the NN function stays the same. To fix this inconsistency, Kwon et al. (2021) proposed adaptive sharpness (invariant under special symmetries), and Andriushchenko et al. (2023) use adaptive notions of sharpness that are invariant to element-wise scaling,

$$S_{\text{max}}^{\text{ad}}(\boldsymbol{w}, \boldsymbol{c}) = \mathbb{E}_{\mathbb{S}}\left[\max_{\|\boldsymbol{\delta} \oslash \boldsymbol{c}\|_2 \leq \rho} L_{\mathbb{S}}(\boldsymbol{\theta} + \boldsymbol{\delta}) - L_{\mathbb{S}}(\boldsymbol{\theta})\right], \quad (1)$$

with scaling vector $\boldsymbol{c}$ (usually set to $|\boldsymbol{\theta}|$, Kwon et al., 2021).

**The problem:** Adaptive sharpness only considers the symmetry induced by element-wise re-scaling. But symmetries of transformers go beyond the invariance that adaptive sharpness captures. Maybe unsurprisingly, Andriushchenko et al. (2023) find inconsistent trends for adaptive sharpness in transformers, with sharpness failing to correlate with generalisation, versus other architectures. We hypothesize this is related to adaptive sharpness not accounting for the full symmetry in transformers. In this paper, we address this. The central question is: *If adaptive sharpness is the fix for a special symmetry, can we provide a more general solution for the symmetries of transformers, to fix the above inconsistency?*

## 3.1. Symmetries in Neural Networks

Here, we give a brief overview and make the notion of NN symmetries more concrete, focusing on those studied by Kunin et al. (2021). Those symmetries lead to rather small effective dimensionality reduction as they are often of $\mathrm{GL}(1)$ or $\mathrm{GL}^+(1)$, but they can still impact the network behaviour. Let $\boldsymbol{\theta}$ denote the parameters of a neural net, $\mathbf{1}_{\mathcal{A}}$ a binary mask, and $\mathbf{1}_{\neg\mathcal{A}}$ its complement such that their sum is a vector of ones, $\mathbf{1}_{\mathcal{A}} + \mathbf{1}_{\neg\mathcal{A}} = \mathbf{1}$. Let $\boldsymbol{\theta}_{\mathcal{A}} := \boldsymbol{\theta} \odot \mathbf{1}_{\mathcal{A}}$ with $\odot$ the element-wise product. Further, let $\mathcal{A}_{1,2}$ be two disjoint subsets, $\mathcal{A}_1 \cap \mathcal{A}_2 = \emptyset$ with masks $\mathbf{1}_{\mathcal{A}_1}, \mathbf{1}_{\mathcal{A}_2}$. Then we have the following common symmetries, characterized by their symmetry group $\mathcal{G}$, such that for any $g \in \mathcal{G}$ the parameters $\psi(g, \boldsymbol{\theta})$ and $\boldsymbol{\theta}$ represent the same function:

- **Translation:** $\psi(\boldsymbol{\alpha}, \boldsymbol{\theta}) = \mathbf{1}_{\mathcal{A}} \odot \boldsymbol{\alpha} + \boldsymbol{\theta}$ with $\boldsymbol{\alpha} \in \mathbb{R}^h$

- **Scaling:** $\psi(\alpha, \boldsymbol{\theta}) = \alpha\boldsymbol{\theta}_{\mathcal{A}} + \boldsymbol{\theta}_{\neg\mathcal{A}}$ with $\alpha \in \mathbb{R}_{>0}$

- **Re-scaling:** $\psi(\alpha, \boldsymbol{\theta}) = \alpha\boldsymbol{\theta}_{\mathcal{A}_1} + 1/\alpha\boldsymbol{\theta}_{\mathcal{A}_2} + \boldsymbol{\theta}_{\neg(\mathcal{A}_1 \vee \mathcal{A}_2)}$ with $\alpha \in \mathbb{R}_{>0}$

Their associated groups are $\mathcal{G} = \mathbb{R}^h, \mathrm{GL}^+(1), \mathrm{GL}^+(1)$. In practice, there may be multiple symmetries acting onto disjoint parameter sub-spaces. Note that re-scaling is essentially the symmetry that adaptive sharpness corrects for.

## 3.2. Rescale Symmetry of Transformers

Transformers exhibit a higher-dimensional symmetry than the previous examples; we formalize the treatment of this symmetry in the following canonical form.

**Definition 3.1** (Functional GL-symmetric building block)**.** Consider a function $\overline{\overline{f}}(\boldsymbol{G}, \boldsymbol{H})$ on $\mathbb{R}^{m \times h} \times \mathbb{R}^{n \times h}$ that consumes two matrices $\boldsymbol{G} \in \mathbb{R}^{n \times h}, \boldsymbol{H} \in \mathbb{R}^{m \times h}$ but only uses the product $\boldsymbol{G}\boldsymbol{H}^\top$, i.e. $\overline{\overline{f}}(\boldsymbol{G}, \boldsymbol{H}) = g(\boldsymbol{G}\boldsymbol{H}^\top)$ for some $g$ over $\mathbb{R}^{m \times n}$. $\overline{\overline{f}}$ is symmetric under the *general linear group*

$$\mathrm{GL}(h) := \left\{ \boldsymbol{A} \in \mathbb{R}^{h \times h} \mid \boldsymbol{A} \text{ invertible} \right\}$$

with $\dim(\mathrm{GL}(h)) = h^2$ and action

$$\psi(\boldsymbol{A}, (\boldsymbol{G}, \boldsymbol{H})) = (\boldsymbol{G}\boldsymbol{A}^{-1}, \boldsymbol{H}\boldsymbol{A}^\top). \qquad (3)$$

In other words, we can insert and then absorb the identity $\boldsymbol{A}^{-1}\boldsymbol{A}$ into $\boldsymbol{G}, \boldsymbol{H}$ to obtain equivalent parameters $\boldsymbol{G}\boldsymbol{A}^{-1}, \boldsymbol{H}\boldsymbol{A}^\top$ that represent the same function.

Example A.2 illustrates GL symmetry for a shallow linear net. Indeed, many popular NN building blocks feature this form, most prominently the attention mechanism in transformers (Vaswani et al., 2017). We give the attention symmetry in Example A.1, and we provide the symmetry for low-rank adapters (Hu et al., 2022) in Example A.3. These examples are NN building blocks that introduce GL symmetries into a loss function and can all be treated through the canonical form in Definition 3.1. In contrast to the symmetries from Section 3.1, they lead to more drastic dimensionality reduction. Consider for example a single self-attention layer where $h = h_\mathrm{v} = h_\mathrm{k}$. The number of trainable parameters is $4h^2$ and the two $\mathrm{GL}(d)$ symmetries reduce the effective dimension to $4h^2 - 2\dim(\mathrm{GL}(h)) = 2h^2$, i.e. they render *half* the parameter space redundant. We hypothesize that the impact of a low-dimensional symmetry on objects like the Euclidean Hessian's trace (Dinh et al., 2017) may be amplified for such higher-dimensional symmetries.

## 3.3. Mathematical Concepts for Riemannian Geometry

We now outline required properties of manifolds for the full development of our approach. We list essential concepts here, and provide definitions and a brief review Appendix B. For further information, see for instance Lee (2003). Figure 2 illustrates the main concepts we will require.

**Ambient embedding space:** We assume that the manifold of possible parameters is embedded in a linear Euclidean space $\mathcal{E} \simeq \mathbb{R}^d$ with $d$ the number of parameters. We can think of $\mathcal{E}$ as the *ambient space*. For instance, for a loss function $\overline{\overline{\ell}} : \mathcal{E} \to \mathbb{R}, \boldsymbol{\theta} \mapsto \overline{\overline{\ell}}(\boldsymbol{\theta})$, we can use ML libraries to evaluate its value, as well as its Euclidean gradient

$$\mathrm{grad}_{\boldsymbol{\theta}} \overline{\overline{\ell}} = \left( \frac{\partial\overline{\overline{\ell}}(\boldsymbol{\theta})}{\partial\theta_i} \right)_{i=1,\dots,d} \in \mathbb{R}^d.$$

Because the geometry of $\mathcal{E}$ is flat, i.e. uses the standard metric $\langle \boldsymbol{\theta}_1, \boldsymbol{\theta}_2 \rangle := \boldsymbol{\theta}_1^\top \boldsymbol{\theta}_2$, this object consists of partial derivatives. However, the Riemannian generalization will add correction terms. In what follows we consider only the restriction of objects like $\overline{\overline{\ell}}$ to the parameter manifold.

**Definition 3.2.** We take $\overline{\mathcal{M}}$ to be the manifold of network parameters, and consider it a sub-manifold embedded into $\mathcal{E}$, the computational space of matrices on which all our numerical calculations are done. We call $\overline{\mathcal{M}}$ the *total space*. On the total space we have a loss function $\overline{\ell} : \overline{\mathcal{M}} \to \mathbb{R}$.

Our goal is to calculate derivatives/geometric quantities after removing the NN's symmetries. The symmetry relation induces natural equivalence classes, which we write $[\boldsymbol{\theta}]$, and explain in Appendix B.1. We let $\mathcal{M} = \overline{\mathcal{M}}/\sim$ represent the *quotient* of the original parameter space manifold by the equivalence relation $\sim$ associated with the symmetry (Appendix B.2). We also require *tangent vectors*; these are straightforward on the total space $\overline{\mathcal{M}}$, but the tangent space of the quotient manifold, $\mathcal{M}$, requires more machinery: *vertical* and *horizontal spaces*, and corresponding *lift*s. These concepts are all defined in Appendix B.3.

Once we endow $\overline{\mathcal{M}}$ with a smooth inner product over its tangent vectors, we obtain a *Riemannian manifold* (defined in Appendix B.4). This construction lets us analyze differ-

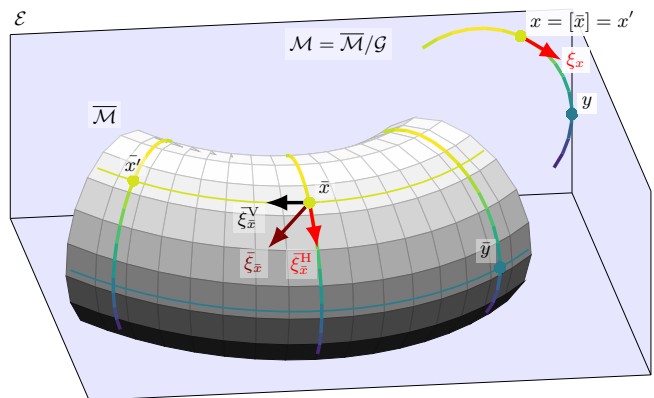

| | |
|---|---|
| $\mathcal{E}$ | Ambient embedding space |
| $\overline{\mathcal{M}}$ | Total space |
| $\mathcal{M}$ | Quotient space |
| $\mathcal{G}$ | Symmetry group |
| $\bar{x}, \bar{y}$ | Points on the total space |
| $x, y$ | Points on the quotient space |
| $\bar{\xi}_{\bar{x}}$ | Tangent vector in the tangent space at point $\bar{x}$, $\mathrm{T}_{\bar{x}}\overline{\mathcal{M}}$ |
| $\xi_x$ | Tangent vector in the tangent space at point $x$, $\mathrm{T}_x\mathcal{M}$ |
| $\bar{\xi}_{\bar{x}}^{\mathrm{V}}$ | Vertical component of $\bar{\xi}_{\bar{x}}$ in the vertical space $\mathcal{V}_{\bar{x}}\overline{\mathcal{M}}$ |
| $\bar{\xi}_{\bar{x}}^{\mathrm{H}}$ | Horizontal component of $\bar{\xi}_{\bar{x}}$ in the horizontal space $\mathcal{H}_{\bar{x}}\overline{\mathcal{M}} \simeq \mathrm{T}_x\mathcal{M}$, horizontal lift of $\xi_x$ |

Figure 2: **Illustrative sketch relating total and quotient space and their tangent spaces.** A tangent vector at a point in total space, $\bar{\xi}_{\bar{x}} \in \mathrm{T}_{\bar{x}}\overline{\mathcal{M}}$ can be decomposed into a horizontal component $\bar{\xi}_{\bar{x}}^{\mathcal{H}}$ and a vertical component $\bar{\xi}_{\bar{x}}^{\mathcal{V}}$. The vertical component points along the direction where the quotient space $x = [\bar{x}]$ remains unaffected. The horizontal component points along the direction that changes the equivalence class. We can use $\bar{\xi}_{\bar{x}}^{\mathcal{H}}$ as a representation of the tangent vector $\xi_x \in T_x\mathcal{M}$ on the quotient space. The component $\bar{\xi}_{\bar{x}}^{\mathcal{H}}$ represents the *horizontal lift* of $\xi_x$.

ential objects that live on quotient manifolds, in the ambient space in a natural way. Furthermore, this allows us to define the horizontal space as the orthogonal complement of the vertical space (Appendix B.4), and to define a *Riemannian gradient* (Appendix B.5). Most properties from the Euclidean case still hold for the Riemannian gradient, but of particular interest to us is the fact that the direction $\mathrm{grad} f(x)$ is still the steepest-ascent direction of $f$ at a point $x$.

We additionally make use of *geodesic curves*. Intuitively, geodesic curves can either be seen as curves of minimal distance between two points on a manifold $\overline{\mathcal{M}}$, or equivalently, as curves through a given point with some initial velocity, and whose acceleration is zero—a generalization of Euclidean straight lines. See Appendix B.6 for details.

Putting it all together, this gives us a *recipe* for computing quantities invariant to a given symmetry relation: (i) find a Riemannian metric compatible with this symmetry; (ii) determine the vertical space for the symmetry relation; (iii) use the metric to find the orthogonal complement of this vertical space, i.e. the projector into the horizontal space; (iv) find the horizontal geodesics. Combined, these steps allow us to do calculations in the quotient manifold along the proper paths (given by geodesics).

## 4. Geodesic Sharpness

We posit that adaptive sharpness measures should take into account the geometry of the quotient parameter manifold that arises after removing symmetries from the parameter space. We base our sharpness measure on the notion of a *geodesic ball*: the set of points that can be reached by geodesics, starting at a point $p$ and whose initial velocity has a norm smaller than $\rho$, after one time unit. In $\mathbb{R}^d$ this

is just the usual definition of a ball, since the geodesics are straight lines. If $\bar{\xi} \in \mathrm{H}_{\bar{x}}\overline{\mathcal{M}}$ is a horizontal vector, and $\bar{\gamma}(t)$ is a geodesic starting at $\theta$ and with initial velocity $\bar{\xi}$:

$$S_{\max}^{\rho}(\boldsymbol{w}) = \mathbb{E}_{\mathbb{S}}\left[\max_{\|\bar{\boldsymbol{\xi}}\|_{\bar{\gamma}(0)} \leq \rho} L_{\mathbb{S}}(\bar{\gamma}_{\bar{\boldsymbol{\xi}}}(1)) - L_{\mathbb{S}}(\bar{\gamma}_{\bar{\boldsymbol{\xi}}}(0))\right]. \quad (4)$$

If the initial velocity, $\bar{\xi}$, is a horizontal vector, then the velocity of the geodesic, $\dot{\bar{\gamma}}_{\bar{\xi}}$, will stay horizontal. The choice of $t = 1$ in $\bar{\gamma}_{\bar{\xi}}(1)$ is not as arbitrary as it first seems (do Carmo, 1992): since for a positive $a$, $\bar{\gamma}_{\bar{\xi}}(at) = \bar{\gamma}_{a\bar{\xi}}(t)$, positions reached with arbitrary $t$ can be reached by instead fixing $t = 1$ and manipulating the initial velocity's norm via $\rho$.

When we do not have an analytical solution for the geodesic, we can use the approximation:

$$\bar{\gamma}_{\bar{\boldsymbol{\xi}}}^i(t) = \bar{\gamma}_{\bar{\boldsymbol{\xi}}}^i(0) + \bar{\xi}^i t - \frac{1}{2}\Gamma_{kl}^i \bar{\xi}^k \bar{\xi}^l t^2 + \mathcal{O}(\bar{\xi}^3), \quad (5)$$

where $\bar{\boldsymbol{\xi}} = (\bar{\xi}^i)$ is the initial (horizontal) velocity, and $\Gamma_{kl}^i$ are the Christoffel symbols. We show that geodesic sharpness reduces to adaptive sharpness measures in Appendix F, under appropriate metric choices and by taking a first-order approximation to the geodesics, that is, ignoring the terms corresponding to the curvature, $\Gamma_{kl}^i$.

## 5. Geodesic Sharpness in Practice

We now apply geodesic sharpness to concrete examples. A fully worked out scalar toy model is in Appendix D.

Following previous works by Dziugaite et al. (2020); Kwon et al. (2021); Andriushchenko et al. (2023), we use the Kendall rank correlation coefficient (Kendall, 1938) to assess the correlation between generalization and sharpness

in the empirical validations of our approach:

$$\tau(\boldsymbol{t}, \boldsymbol{s}) = \frac{2}{M(M-1)} \sum_{i<j} \text{sign}(t_i - t_j)\,\text{sign}(s_i - s_j)\,,$$

where $\boldsymbol{t}$ and $\boldsymbol{s}$ are the vectors of observed variables between which we are measuring correlation.

Although the criterion of symmetry compatibility restricts the class of suitable metrics, these are not necessarily unique. As long as it is symmetry-compatible, we have no reason to prefer one metric over another, except for practical aspects like numerical cost and stability. We will present results on two symmetry-compatible metrics that are simple, yet non-trivial, and often used in the related literature on Riemannian optimization on fixed-rank matrix spaces (Luo et al., 2023).

### 5.1. Diagonal Networks

We start by studying *diagonal linear nets*, one of the simplest non-trivial neural networks (Pesme et al. (2021), Woodworth et al. (2020)). These have two parameters, $\boldsymbol{u}, \boldsymbol{v}$, and predict a label, $y$, given an input, $\boldsymbol{x}$, via $y = \boldsymbol{x}^\top(\boldsymbol{u} \odot \boldsymbol{v})$. We consider linear regression with labels $y \in \mathbb{R}^n$, a data matrix $\boldsymbol{X} \in \mathbb{R}^{n \times d}$, and take as our loss $L(\boldsymbol{u}, \boldsymbol{v}) = \|\boldsymbol{X}(\boldsymbol{u} \odot \boldsymbol{v}) - y\|_2^2$. Our parameter manifold $\overline{\mathcal{M}}$ is $\mathbb{R}^d \times \mathbb{R}^d$.

The nets are symmetric under element-wise rescaling: $(\boldsymbol{u}, \boldsymbol{v}) \mapsto (\alpha\boldsymbol{u}, \alpha^{-1}\boldsymbol{v})$, leaves $\boldsymbol{\beta} = \boldsymbol{u} \odot \boldsymbol{v}$ and $L$ invariant.

**Metric:** At a point $(\boldsymbol{u}, \boldsymbol{v}) \in \overline{\mathcal{M}}$, for two tangent vectors $\boldsymbol{\eta} = (\eta_{\boldsymbol{u}}, \eta_{\boldsymbol{v}})$, $\boldsymbol{\nu} = (\nu_{\boldsymbol{u}}, \nu_{\boldsymbol{v}}) \in T_{(\boldsymbol{u},\boldsymbol{v})}\overline{\mathcal{M}}$, we use the following two symmetry-compatible metrics:

$$\langle \boldsymbol{\eta}, \boldsymbol{\nu} \rangle^{\text{inv}} := \sum_{i=1}^d \frac{\eta_{\boldsymbol{u}}^i \nu_{\boldsymbol{u}}^i}{(\boldsymbol{u}^i)^2} + \frac{\eta_{\boldsymbol{v}}^i \nu_{\boldsymbol{v}}^i}{(\boldsymbol{v}^i)^2}, \tag{6}$$

$$\langle \boldsymbol{\eta}, \boldsymbol{\nu} \rangle^{\text{mix}} := \sum_{i=1}^d \eta_{\boldsymbol{u}}^i \nu_{\boldsymbol{u}}^i (\boldsymbol{v}^i)^2 + \eta_{\boldsymbol{v}}^i \nu_{\boldsymbol{v}}^i (\boldsymbol{u}^i)^2 \,. \tag{7}$$

**Horizontal space:** Both have the same horizontal space

$$\mathcal{H}_{(\boldsymbol{u},\boldsymbol{v})}\overline{\mathcal{M}} = \left\{ (\eta_{\boldsymbol{u}}, \eta_{\boldsymbol{v}}) \in T_{(\boldsymbol{u},\boldsymbol{v})}\mathcal{M} \mid \frac{\eta_{\boldsymbol{u}}^i}{\boldsymbol{u}^i} = \frac{\eta_{\boldsymbol{v}}^i}{\boldsymbol{v}^i} \ \forall i \right\}\,.$$

**Geodesics:** With $\boldsymbol{b}_i := \frac{\eta_{\boldsymbol{u}}^i}{\boldsymbol{u}^i} = \frac{\eta_{\boldsymbol{v}}^i}{\boldsymbol{v}^i}$, the geodesics are

$$\gamma_{\text{inv}}(t)^i = \left(\boldsymbol{u}_0^i \exp(\boldsymbol{b}_i t), \boldsymbol{v}_0^i \exp(\boldsymbol{b}_i t)\right)\,,$$

$$\gamma_{\text{mix}}(t)^i = \left(\boldsymbol{u}_0^i \sqrt{1 + 2\boldsymbol{b}_i t}, \boldsymbol{v}_0^i \sqrt{1 + 2\boldsymbol{b}_i t}\right)\,,$$

with starting points $\boldsymbol{u}_0^i$ and $\boldsymbol{v}_0^i$, i.e. the trained parameters.

**Geodesic sharpness:** Assume that $\boldsymbol{X}^\top \boldsymbol{X} = \boldsymbol{I}_d$ (Andriushchenko et al., 2023), and denote $\boldsymbol{\beta}_0 = \boldsymbol{u}_0 \odot \boldsymbol{v}_0$.

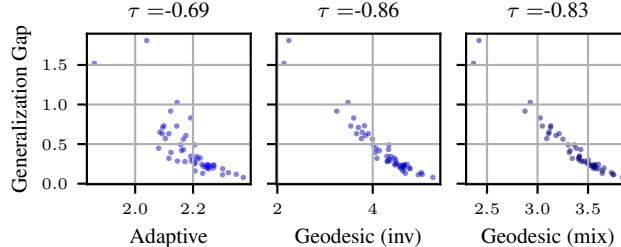

Figure 3. **Adaptive vs. geodesic sharpness on diagonal nets.** The generalization gap is the test loss (remember all models are trained to $10^{-5}$ training loss). The correlation coefficient's magnitude is larger for geodesic sharpness.

The minimum norm least squares predictor is $\boldsymbol{\beta}_* := (\boldsymbol{X}^\top \boldsymbol{X})^{-1} \boldsymbol{X}^\top \boldsymbol{y} = \boldsymbol{X}^\top \boldsymbol{y}$. Using Equation (4) (details in Appendix E), we get (to first and to second order)

$$\begin{aligned} S_{\text{max; inv}}^\rho(\boldsymbol{u}, \boldsymbol{v}) =& 4\rho\|\boldsymbol{\beta}_0 \odot (\boldsymbol{\beta}_0 - \boldsymbol{\beta}_*)\|_2 \\ &+ 4\rho^2 \max\left[(\beta_0^i)^2\right]\,, \end{aligned} \tag{8}$$

which depends on $\rho$ and the difference between the learned, and the optimal minimum norm, predictor. Eq. 8 is the square of adaptive sharpness (when the residual $\|\boldsymbol{\beta}_0 \odot (\boldsymbol{\beta}_0 - \boldsymbol{\beta}_*)\|_2$ is small) if very carefully chosen hyperparameters were used (by contrast, this result naturally appears using our geodesic approach). For the second metric, we have

$$S_{\text{max; mix}}^\rho(\boldsymbol{u}, \boldsymbol{v}) = \rho\|\boldsymbol{\beta}_0 - \boldsymbol{\beta}_*\|_2\,.$$

#### 5.1.1. EMPIRICAL VALIDATION

**Experimental setup:** We follow Andriushchenko et al. (2023), generate a randomly distributed data matrix $\boldsymbol{X}$, a random ground-truth vector $\boldsymbol{\beta}^*$ that is 90% sparse, and train 50 diagonal networks to $10^{-5}$ training loss on a regression task.

We focus on the more practically relevant case of overparametrization ($d > n$). One downside of this is that the theoretical expressions derived in the previous section, while a useful sanity check, no longer hold (since overparameterization breaks the assumption $\boldsymbol{X}^\top \boldsymbol{X} = \boldsymbol{I}_{d=200}$). To obtain our geometric sharpness, we directly solve Equation (4).

**Results:** All three notions of sharpness are able to predict generalization (Figure 3). Geodesic sharpness, although closely related for diagonal nets to adaptive worst-case sharpness, does slightly better. This applies to both metrics studied, and they perform roughly the same. See Section 7 for comments about the sign of the correlation.

### 5.2. Attention Layers

Next, we look at the symmetric functional block from Definition 3.1. Our computation space is $\mathcal{E} := \mathbb{R}^{n \times h} \times \mathbb{R}^{m \times h} \simeq$

$\mathbb{R}^{(n+m)h}$ and we restrict weights to have full column rank:

**Assumption 5.1.** The rank of $G, H$ corresponds to their number of columns, $\text{rank}(G) = \text{rank}(H) = h$.

This implies $h \leq n, m$, which is usually satisfied in (multi-head) attention layers (Example A.1) for the default choices of $d_\text{v}, d_\text{k}$. While the weights of multi-head attention tend to have high column rank (Yu & Wu, 2023), they are not guaranteed to be full column rank. To account for this, we introduce a small relaxation parameter, $\epsilon$, to the Gram matrices s.t. $G^\top G \to G^\top G + \epsilon I_h$. Empirically, we observe that as long as $\epsilon$ is sufficiently small, it does not affect our results (Appendix H.2). Therefore, we restrict both $G, H$ to the set of fixed-rank matrices, $\overline{\mathcal{M}} \leftarrow \mathbb{R}_h^{n \times h} \times \mathbb{R}_h^{m \times h}$ where $\mathbb{R}_k^{n \times h} := \{ B \in \mathbb{R}^{n \times h} \mid \text{rank}(B) = k \}$. We can represent a point $\bar{x} \in \overline{\mathcal{M}}$ by a matrix tuple $(G, H) \in \mathbb{R}_h^{n \times h} \times \mathbb{R}_h^{m \times h}$. Its tangent space $\text{T}_{\bar{x}}\overline{\mathcal{M}}$ is

$$\text{T}_{\bar{x}}\overline{\mathcal{M}} = \left\{ \bar{\eta} = (\bar{\eta}_G, \bar{\eta}_H) \in \mathbb{R}^{n \times h} \times \mathbb{R}^{m \times h} \right\},$$

**Metric:** We endow $\overline{\mathcal{M}}$ with the two metrics $\langle \cdot, \cdot \rangle_{\bar{x}}^{\text{inv,mix}}$ : $\text{T}_{\bar{x}}\overline{\mathcal{M}} \times \text{T}_{\bar{x}}\overline{\mathcal{M}} \to \mathbb{R}$ (proof they are valid in Appendix I.1):

$$\langle \bar{\eta}, \bar{\zeta} \rangle_{\bar{x}}^{\text{inv}} := \text{Tr} \left( (G^\top G)^{-1} \bar{\eta}_G^\top \bar{\zeta}_G + (H^\top H)^{-1} \bar{\eta}_H^\top \bar{\zeta}_H \right), \tag{9}$$

$$\langle \bar{\eta}, \bar{\zeta} \rangle_{\bar{x}}^{\text{mix}} := \text{Tr} \left( (H^\top H) \bar{\eta}_G^\top \bar{\zeta}_G + (G^\top G) \bar{\eta}_H^\top \bar{\zeta}_H \right). \tag{10}$$

They differ from the Euclidean metric that simply flattens and concatenates the matrix tuples into vectors and takes their dot product, $\langle \eta, \zeta \rangle = \text{Tr} \left( \eta_G^\top \zeta_G + \eta_H^\top \zeta_H \right)$. Importantly, they are invariant under symmetries of the attention mechanism, and thus define valid metrics on the quotient manifold (Absil et al., 2008).

**Horizontal space:** For $\langle \cdot, \cdot \rangle_{\bar{x}}^{\text{inv, mix}}$ and $\bar{\xi}_{G,H} \in \mathbb{R}^{m \times r}$ we have (for a proof, see for example Luo et al. (2023))

$$\mathcal{H}_{\bar{x}}^{\text{inv}}\overline{\mathcal{M}} = \{ (\bar{\xi}_G, \bar{\xi}_H) \mid \bar{\xi}_G^\top G H^\top H = G^\top G H^\top \xi_H^\top \},$$
$$\mathcal{H}_{\bar{x}}^{\text{mix}}\overline{\mathcal{M}} = \{ (\bar{\xi}_G, \bar{\xi}_H) \mid G^\top \bar{\xi}_G H^\top H = G^T G \xi_H^\top H \}.$$

**Projection onto horizontal space:** Given $\xi \in \text{T}_{\bar{x}}\overline{\mathcal{M}}$ in the total tangent space, the horizontal space is

$$\mathcal{H}_{\bar{x}}^{\text{inv, mix}}\overline{\mathcal{M}} = \left\{ (\bar{\xi}_G + G\Lambda^{\text{inv, mix}}, \bar{\xi}_H - H(\Lambda^{\text{inv, mix}})^\top) \right\}$$

where $\Lambda^{\text{inv}}$ solves the Sylvester equation $A\Lambda + \Lambda A^\top = B$, with $A = G^\top G H^\top H$, $B = G^\top G H^\top \bar{\xi}_H - \bar{\xi}_G^\top G H^\top H$, whereas $\Lambda^{\text{mix}}$ has an explicit form: $\Lambda^{\text{mix}} = {}^1\!/{}_2 \left( \bar{\xi}_H^\top H (H^\top H)^{-1} - (G^\top G)^{-1} G^\top \bar{\xi}_G \right)$.

**Geodesics:** We are unaware of analytical solutions for the geodesics of either (Eq. 9 and Eq. 10), so we approximate

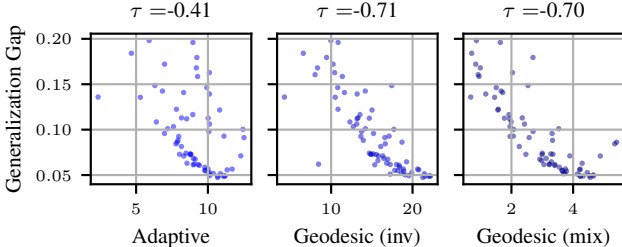

$\tau = $-0.41 $\qquad$ $\tau = $-0.71 $\qquad$ $\tau = $-0.70

Generalization Gap

Adaptive $\qquad$ Geodesic (inv) $\qquad$ Geodesic (mix)

**Figure 4: Adaptive vs. geodesic sharpness on ImageNet ViTs.** We use 72 trained models from Wortsman et al. (2022), and measure their generalization gap as the difference between test and train error. The correlation coefficient's magnitude is larger for geodesic sharpness.

them with Eq. 5. For horizontal tangent vectors $(\bar{\xi}_G, \bar{\xi}_H)$, we have for $\langle \cdot, \cdot \rangle_{\bar{x}}^{\text{inv}}$

$$(\Gamma_{kl}^i)^{\text{inv}} \bar{\xi}_G^k \bar{\xi}_G^l = - \bar{\xi}_G (G^\top G)^{-1} \left[ \bar{\xi}_G^\top G + G^\top \bar{\xi}_G \right] + G(G^\top G)^{-1} \bar{\xi}_G^\top \bar{\xi}_G \tag{11}$$

(similar for the $H$ components). For $\langle \cdot, \cdot \rangle_{\bar{x}}^{\text{mix}}$, the geodesic equations are coupled and the $G$ components are

$$\left[ (\Gamma_{kl}^i)^{\text{mix}} \bar{\xi}^k \bar{\xi}^l \right]_G = \bar{\xi}_G \left[ \bar{\xi}_H^\top H + H^\top \bar{\xi}_H \right] (H^\top H)^{-1} - G(\bar{\xi}_H^T \bar{\xi}_H)(H^\top H)^{-1} \tag{12}$$

(the $H$ components are similar, proof in Appendix I.2).

### 5.3. Transformers

Transformers have a mix of attention layers and layers with more restricted symmetries for which adaptive sharpness is more appropriate. We present in Appendix C.1 how we treat each layer of transformers. We introduce relaxations In Appendix C.2 we present Algorithm 1, which we use to solve for geodesic sharpness.

#### 5.3.1. EMPIRICAL VALIDATION: VISION TRANSFORMERS

**Experimental setup:** We follow Andriushchenko et al. (2023), and look at models obtained from fine-tuning CLIP on ImageNet-1k (Radford et al., 2021). Specifically, we use the trained classifiers after fine-tuning a CLIP ViT-B/32 on ImageNet with randomly selected hyperparameters from (Wortsman et al., 2022). We compute adaptive worst-case, and our geodesic, sharpness on the same 2048 data points from the ImageNet training set, divided into batches of 256, by calculating sharpness on each batch separately, then averaging the results. The generalization gap is the difference between test and training error.

**Results:** Figure 4 shows our results. We find a strong correlation between geodesic sharpness and the generalization gap on ImageNet. This correlation is stronger than

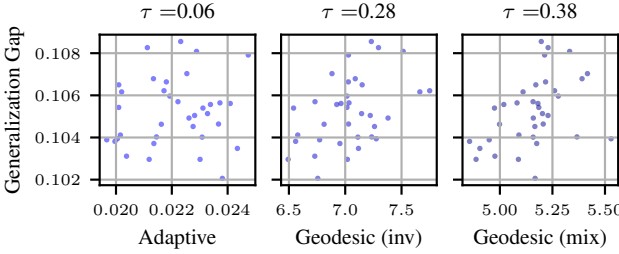

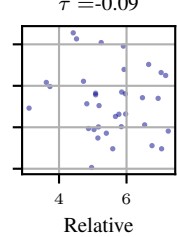

Figure 6: **Extension of Figure 5 to relative sharpness.** We find that relative flatness (Petzka et al., 2021) fails to find a significant correlation, compared to our geodesic sharpness.

Figure 5: **Adaptive vs. geodesic sharpness on MNLI language models.** We use 35 trained models from McCoy et al. (2020), and show the generalization gap on the MNLI dev matched set (Williams et al., 2018). Geodesic sharpness shows the largest correlation.

that observed with adaptive sharpness and is consistently negative, implying that the geodesically sharpest models studied on ImageNet are those that generalize best–contrary to what might have been expected, but consistent with the correlation from the diagonal networks.

### 5.3.2. EMPIRICAL VALIDATION: LANGUAGE MODELS

**Experimental Setup:** We also consider BERT models that were fine-tuned on MNLI (Williams et al., 2018) by Mc-Coy et al. (2020) . We compute adaptive worst-case, and our geodesic, sharpness on the same 1024 data points from the MNLI training set, with batches of 128 points, by calculating then averaging sharpness on each batch.

**Results:** Figure 5 shows our results. We find a consistent correlation between geodesic sharpness and the generalization gap on MNLI for both metrics, while adaptive sharpness ($\tau = 0.06$) cannot find any correlation. The correlation is positive, i.e. geodesically flatter models generalize better.

## 6. Additional Experiments

### 6.1. Comparison With Relative Sharpness

Relative sharpness (Petzka et al., 2021) is a promising sharpness measure that has proven useful in regularizing transformer training, outperforming other approaches (Adilova et al., 2023). We compare it with our geodesic sharpness in the language model setting from Section 5.3.2; see Figure 6.

### 6.2. Verification of Reparametrization Invariance

Mathematically, geodesic sharpness is invariant to symmetry transformations of the form of Equation (3). Here, we verify empirically that our practical version that can be computed efficiently numerically is close to invariant.

**Experimental setup:** We take a single batch and language model from Section 5.3.2, and compute geodesic sharpness

for various points on an orbit that represent the same function. Specifically, we reparametrize using $\boldsymbol{A} = a\boldsymbol{G}$, where $\boldsymbol{G}$ is a random standard Gaussian matrix (which is almost always invertible and sampled once in each run), and control the scale $a$. We sample one $\boldsymbol{G}$ for each attention head. We compare this with adaptive sharpness.

**Results:** Figure 7 visualizes the sharpness ratio before and after reparameterization. The colors represent different values of the scale factor, which goes from $10^{-2}$ to $10^{2}$. Our numerically computed geodesic sharpness remains constant.

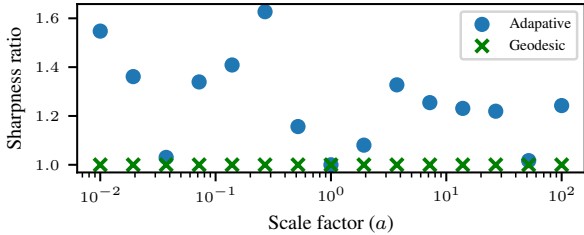

Figure 7: **Variation of adaptive vs. geodesic sharpness within an orbit where the neural net function remains unchanged.** We show the ratios between the original sharpness and the sharpness obtained after applying a symmetry transformation. Geodesic sharpness stays constant, whereas adaptive sharpness assumes several different values.

## 7. Remarks, Limitations & Future Work

**Discovering correlation:** Adaptive sharpness, as discussed thoroughly by Andriushchenko et al. (2023), is unable to reveal a correlation between sharpness and generalization for transformers. Our geodesic sharpness consistently recovers strong correlation on transformers, and strengthens the correlation in the case of diagonal networks.

**Metric choice:** Our results are robust w.r.t. the choice of metric, as long as *it captures the parameter symmetry*. The mixed metric yields slightly better results on BERT, perhaps owing to its more stable numerics (e.g. possible inversion of nearly singular matrices is side-stepped). Additionally, the mixed metric avoids calling expensive Sylvester equation solvers and has a simple horizontal space projection.

**Sign of the correlation:** One of our surprising results is that the sign of the correlation between geodesic sharpness and generalization varies depending on the setting and is at times negative, somewhat at odds with the common view that sharpness always *positively* correlates with generalization (i.e., flatter models generalize better). This artifact is not inherent to our proposed metrics. E.g., adaptive sharpness anti-correlates with generalization in our diagonal network setting, but was previously found to positively correlate with generalization on other tasks (Kwon et al., 2021).

Our geodesic sharpness improves over adaptive sharpness in the following sense: Where adaptive sharpness finds no correlation, our metrics do find a signed correlation, and where adaptive sharpness finds signed correlation, our metrics find a stronger similarly-signed correlation. That is, we for the first time observe *consistent correlations* within-task for transformers, opening questions for further study.

**Limitations:** While our *geodesic sharpness* is more general than previous measures, there remain symmetries for which taking the quotient may be computationally expensive or intractable. Still, we think that accounting for some symmetry is better than none, and even under computational constraints it could be useful as a diagnostic "probe".

Our new measures detect previously undetected correlation with generalization. In the process, however, we also discovered that the sign of the correlation, while consistent across metrics and models, can vary across tasks. Until this new variability is understood, this limits the utility of geodesic sharpness, e.g. for regularizing transformer training.

**Future work:** Our work is concerned with accounting for parameter space symmetries that are data-independent. This opens up the question: what is the role of data and how can it be integrated into our framework? A more complete understanding of the interplay between data and parameter symmetries might help explain when geodesic sharpness correlates or anti-correlates with generalization.

## 8. Conclusion

In this paper, we revisited the limitations of traditional sharpness measures attempting to predict generalization for transformers, highlighting how traditional sharpness measures fail to properly account for the rich $GL(h)$ symmetries present in transformers. Addressing this, we introduced geodesic sharpness, a measure defined on the Riemannian quotient manifold obtained by quotienting out transformer symmetries. This framework provides a principled, symmetry-aware measure of sharpness and contains prior adaptive sharpness metrics as first-order approximations.

Through experiments on diagonal networks, vision trans-

formers (ImageNet), and language models (MNLI), we demonstrated that properly accounting for the transformer symmetries restores the correlation between sharpness and generalization. Interestingly, our findings indicate that the sign of the correlation between sharpness and generalization can vary across tasks, suggesting deeper underlying relationships involving data distribution and model structure. This work lays the groundwork for further exploration of these interactions and motivates future research into geometry-informed optimization strategies tailored to transformers.

## Impact Statement

This paper presents work whose goal is to advance the study of deep learning. There are potential indirect societal consequences of our work, none which we feel must be specifically highlighted here.

## Acknowledgements

We would like to express our sincere gratitude to Agustinus Kristiadi and Rob Brekelmans for early feedback on the manuscript. Resources used in preparing this research were provided, in part, by NSERC, the Province of Ontario, the Government of Canada through CIFAR, and companies sponsoring the Vector Institute.

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

# Hide & Seek: Transformer Symmetries Obscure Sharpness & Riemannian Geometry Finds It  (Supplemental Material)

We provide in Table 1 a summary of correlation coefficients between sharpness and generalization for our experiments.

| | Rank correlation coefficient $\tau$ | | |
|---|---|---|---|
| Setting | Adaptive sharpness | $\langle \cdot, \cdot \rangle^{\text{inv}}$ - geodesic sharpness | $\langle \cdot, \cdot \rangle^{\text{mix}}$ - geodesic sharpness () |
| Diagonal networks | -0.68 | -0.83 | **-0.86** |
| ImageNet | -0.41 | **-0.71** | -0.7 |
| MNLI | 0.06 | 0.28 | **0.38** |

Table 1: Summary of the correlation between sharpness measures and generalization. We boldface the best performing metric

In the sections that follow, we provide additional details to supplement the main text.

## A. Additional Examples of GL symmetries Symmetries in Neural Networks

**Example A.1** (Self-attention (Vaswani et al., 2017)). Given a sequence $\boldsymbol{X} \in \mathbb{R}^{t \times d}$ with $t$ tokens and model dimension $d$, self-attention (SA) uses four matrices $\boldsymbol{W}_{\text{q}}, \boldsymbol{W}_{\text{k}} \in \mathbb{R}^{d \times d_{\text{k}}}, \boldsymbol{W}_{\text{v}}, \boldsymbol{W}_{\text{o}}^{\top} \in \mathbb{R}^{d \times d_{\text{v}}}$ (usually, $d = d_{\text{v}} = d_{\text{k}}$) to produce a new $t \times d$ sequence

$$
\begin{aligned}
&\text{SA}(\boldsymbol{W}_{\text{q}}, \boldsymbol{W}_{\text{k}}, \boldsymbol{W}_{\text{v}}, \boldsymbol{W}_{\text{o}}) \\
&= \text{softmax}\left( \frac{\boldsymbol{X} \boldsymbol{W}_{\text{q}} \boldsymbol{W}_{\text{k}}^{\top} \boldsymbol{X}^{\top}}{\sqrt{d_{\text{k}}}} \right) \boldsymbol{X} \boldsymbol{W}_{\text{v}} \boldsymbol{W}_{\text{o}} \,.
\end{aligned}
\tag{13}
$$

This block contains two GL symmetries: one of dimension $d_{\text{k}}$ between the key and query projection weights, $\boldsymbol{G}, \boldsymbol{H} \leftarrow \boldsymbol{W}_{\text{q}}, \boldsymbol{W}_{\text{k}}$, and one of dimension $d_{\text{v}}$ between the value and out projection weights, $\boldsymbol{G}, \boldsymbol{H} \leftarrow \boldsymbol{W}_{\text{v}}, \boldsymbol{W}_{\text{o}}^{\top}$. Similar to Eq. 14, we can account for biases in the key, query, and value projections by appending them to their weight,

$$
\boldsymbol{G}, \boldsymbol{H} \leftarrow \begin{pmatrix} \boldsymbol{W}_{\text{k}} \\ \boldsymbol{b}_{\text{k}}^{\top} \end{pmatrix}, \begin{pmatrix} \boldsymbol{W}_{\text{q}} \\ \boldsymbol{b}_{\text{q}} \end{pmatrix}^{\top} , \quad \boldsymbol{G}, \boldsymbol{H} \leftarrow \begin{pmatrix} \boldsymbol{W}_{\text{v}} \\ \boldsymbol{b}_{\text{v}} \end{pmatrix}, \boldsymbol{W}_{\text{o}}^{\top} \,.
$$

Commonly, $H$ attention heads $\{\boldsymbol{W}_{\text{q}}^{i}, \boldsymbol{W}_{\text{k}}^{i}, \boldsymbol{W}_{\text{v},i}^{i}, \boldsymbol{W}_{\text{o}}^{i}\}_{i=1}^{H}$ independently process $\boldsymbol{X}$ and concatenate their results into the final output (usually $d_{\text{k}} = d_{\text{v}} = d/H$). This introduces $2H$ GL symmetries. Everything also applies to general attention where, instead of $\boldsymbol{X}$, independent data is fed as keys, queries, and values to Eq. 13.

**Example A.2** (Shallow linear net). Consider a two-layer linear net $\text{NN}(\boldsymbol{W}_2, \boldsymbol{W}_1) = \boldsymbol{W}_2 \boldsymbol{W}_1 \boldsymbol{x}$ with weight matrices $\boldsymbol{W}_1 \in \mathbb{R}^{h \times d_{\text{in}}}, \boldsymbol{W}_2 \in \mathbb{R}^{d_{\text{out}} \times h}$ and some input $\boldsymbol{x} \in \mathbb{R}^{d_{\text{in}}}$. This net has GL symmetry with correspondence $\boldsymbol{G}, \boldsymbol{H} \leftarrow \boldsymbol{W}_2, \boldsymbol{W}_1^{\top}$ to Definition 3.1. With first-layer bias, we have

$$
\boldsymbol{W}_2(\boldsymbol{W}_1 \boldsymbol{x} + \boldsymbol{b}_1) = \boldsymbol{W}_2 \begin{pmatrix} \boldsymbol{W}_1 & \boldsymbol{b}_1 \end{pmatrix} \begin{pmatrix} \boldsymbol{x} \\ 1 \end{pmatrix} ,
\tag{14}
$$

corresponding to $\boldsymbol{G}, \boldsymbol{H} \leftarrow \boldsymbol{W}_2, \begin{pmatrix} \boldsymbol{W}_1 & \boldsymbol{b}_1 \end{pmatrix}^{\top}$.

**Example A.3** (Low-rank adapters (LoRA, Hu et al. (2022))). Fine-tuning tasks with large language models add a trainable low-rank perturbation $\boldsymbol{L} \in \mathbb{R}^{d_1 \times h}, \boldsymbol{R} \in \mathbb{R}^{d_2 \times h}$ to the pre-trained weight $\boldsymbol{W} \in \mathbb{R}^{d_1 \times d_2}$,

$$
\text{LoRA}(\boldsymbol{W}) = \boldsymbol{W} + \boldsymbol{L} \boldsymbol{R}^{\top} ,
\tag{15}
$$

introducing a $\mathrm{GL}(h)$ symmetry where $\boldsymbol{G}, \boldsymbol{H} \leftarrow \boldsymbol{L}, \boldsymbol{R}$. Yen et al. (2024) propose an invariant way to train the parameters $\boldsymbol{L}, \boldsymbol{R}$ and show that doing so improves the result obtained via LoRA.

# B. Concepts and Review for Riemannian Geometry

Recall that $\overline{\mathcal{M}}$ is the total space: the manifold of parameters of our network. Also, on the total space we have a loss function $\ell : \overline{\mathcal{M}} \to \mathbb{R}$. Useful resources are Lee (2003), Absil et al. (2008), and Boumal (2023).

### B.1. Orbit of $x$

A symmetry relation naturally defines an equivalence relation: two points $x, y \in \overline{\mathcal{M}}$ are equivalent under the symmetry, if they can be mapped onto each other by the action,

$$ x \sim y \quad \Leftrightarrow \quad \exists g \in \mathcal{G} : y = \psi(g, x) \,. \tag{16} $$

In other words, if we let $\mathrm{orbit}(x) \coloneqq \{\psi(g, x) \mid g \in \mathcal{G}\}$ be all points on the total space that are reachable from $x$ through the action of $\mathcal{G}$, all points in an orbit are equivalent. Instead of $\mathrm{orbit}(x)$, we will write

$$ [x] \coloneqq \{y \in \overline{\mathcal{M}} \mid y \sim x\} \tag{17} $$

for the symmetry-induced equivalence class $[x]$ of $x \in \overline{\mathcal{M}}$.

Let's further assume that $\overline{\ell}$ is symmetric under $\mathcal{G}$, i.e. for any $x \in \overline{\mathcal{M}}$ and all $g \in \mathcal{G}$, $\overline{\ell}(x) = \overline{\ell}(\psi(g, x))$.

### B.2. Quotient $\mathcal{M}$ and Natural Projection

If we take the quotient of the original parameter space manifold $\overline{\mathcal{M}}$, by the equivalence relation, $\sim$, induced by the symmetries of our neural architecture, we get a quotient $\mathcal{M} = \overline{\mathcal{M}}/\sim$. Under certain conditions, $\mathcal{M}$ is a quotient manifold. The mapping between a point in total space to its equivalence class is called the natural projection:

**Definition B.1.** Let $\pi : \overline{\mathcal{M}} \to \overline{\mathcal{M}}/\sim$, be defined by $\overline{x} \mapsto x$. $\pi$ is called the natural, or canonical projection. We use $\pi(\overline{x})$ to denote $x$ viewed as a point of $\mathcal{M} \coloneqq \overline{\mathcal{M}}/\sim$.

### B.3. Tangent Space, Vertical and Horizontal Spaces

Tangent vectors on the total space $\overline{\mathcal{M}}$, embedded in a vector space $\mathcal{E}$ can be viewed as tangent vectors to $\mathcal{E}$, but the tangent space of the quotient manifold, $\mathcal{M}$ is not as straightforward. First, note that any element $\overline{\xi} \in T_{\overline{x}}\mathcal{M}$ that satisfies $D\pi(\overline{x})[\overline{\xi}] = \xi$ (where $D$ is the differential) is a candidate for a representation of $\xi \in T_x\mathcal{M}$. These aren't unique, and as we wish to work without any numerical ambiguity we introduce the notions of the vertical and horizontal spaces:

**Definition B.2.** For a quotient manifold $\mathcal{M} = \mathcal{M}/\sim$, the vertical space at $\overline{x} \in \mathcal{M}$ is the subspace $V_{\overline{x}} = T_{\overline{x}}\mathcal{F} = \ker D\pi(x)$ where $\mathcal{F} = \{\overline{y} \in \mathcal{M} : \overline{y} \sim \overline{x}\}$ is the fiber of $\overline{x}$. The complement of $V_{\overline{x}}$ is the horizontal space at $\overline{x}$: $T_{\overline{x}}\overline{\mathcal{M}} = V_{\overline{x}} \oplus H_{\overline{x}}$.

**Definition B.3.** There is only one element $\overline{\xi}_{\overline{x}}$ that belongs to $H_{\overline{x}}$ and satisfies $D\pi(\overline{x})[\overline{\xi}_{\overline{x}}] = \xi$. This unique vector is called the *horizontal lift* of of $\xi$ at $\overline{x}$. We denote the operator that affects the procedure by $\mathrm{lift}_{\overline{x}}(\cdot)$ When the ambient space, $\mathcal{E}$ is a subset of $\mathbb{R}^{n \times p}$, the horizontal space can also be seen as such a subset, providing a convenient matrix representation of *a priori* abstract tangent vectors of $\mathcal{M}$.

### B.4. Riemannian Manifold

We give our total space $\overline{\mathcal{M}}$ a smooth inner product over its tangent vectors to give a Riemannian manifold.

**Definition B.4.** A Riemannian manifold is a pair $(\mathcal{M}, g)$, where $\mathcal{M}$ is a smooth manifold and $g$ is a Riemannian metric, defined as the inner product on the tangent space $T_x\mathcal{M}$ for each point $x \in \mathcal{M}$, $g_x(\cdot, \cdot) : T_x\mathcal{M} \times T_x\mathcal{M} \to \mathbb{R}$. We also use the notation $\langle \cdot, \cdot \rangle_x$ to denote the inner product.

Note that this definition is not as arcane as it may appear since any smooth manifold admits a Riemannian metric, and we can consider the space of parameters of most neural architectures as constituting a smooth manifold, admitting at least a simple, Euclidean, metric.

The horizontal space can now be defined as the *orthogonal* complement of the vertical space: $H_{\bar{x}} = (V_{\bar{x}})^{\perp} = \{u \in T_{\bar{x}}\overline{\mathcal{M}} : \langle u, v \rangle_x = 0$ for all $v \in V_{\bar{x}}\}$. Additionally, letting $\bar{g}_{\bar{x}}$ denote the metric on $\overline{\mathcal{M}}$, if for every $x \in \mathcal{M}$ and every $\xi_x, \zeta_x$ in $T_x\mathcal{M}$, $\bar{g}_{\bar{x}}(\bar{\xi}_{\bar{x}}, \bar{\zeta}_{\bar{x}})$ does not depend on $\bar{x} \in \pi^{-1}(x)$ then, $g_x(\xi_x, \zeta_x) = \bar{g}_{\bar{x}}(\bar{\xi}_{\bar{x}}, \bar{\zeta}_{\bar{x}})$ defines a valid metric on the quotient manifold $\mathcal{M}$.

### B.5. Riemannian Gradient

**Definition B.5.** If $\bar{f}$ is a smooth scalar field on a Riemannian manifold $\overline{\mathcal{M}}$, then the *gradient* of $\bar{f}$ at $\bar{x}$, $\text{grad}\bar{f}(\bar{x})$ is the unique element of $T_{\bar{x}}\overline{\mathcal{M}}$ such that

$$\langle \text{grad}\bar{f}(\bar{x}), \bar{\xi} \rangle_{\bar{x}} = D\bar{f}(\bar{x})[\bar{\xi}], \forall \bar{\xi} \in T_{\bar{x}}\overline{\mathcal{M}}$$

If $\bar{f}$ is a function on $\overline{\mathcal{M}}$, that induces a function $f$ on a quotient manifold $\mathcal{M}$ of $\overline{\mathcal{M}}$, then we can express the horizontal lift of grad $f$ at $\bar{x}$ as

$$\text{lift}_{\bar{x}}(\text{grad f}) = \text{grad}\bar{f}(\bar{x}).$$

### B.6. Geodesic Curves

**Definition B.6.**

(a) Geodesic curves, $\bar{\gamma}$, are the curves of minimal distance between two points on a manifold $\overline{\mathcal{M}}$. The distance along the geodesic is called the *geodesic distance*. If $\mathcal{M}$ is a Riemannian quotient manifold of $\overline{\mathcal{M}}$, with canonical projection $\pi$, and $\bar{\gamma}$ is a geodesic on $\overline{\mathcal{M}}$, then $\gamma = \pi \circ \bar{\gamma}$ is a geodesic curve on $\mathcal{M}$.

(b) Alternatively, geodesics, $\bar{\gamma}(t) = 0$ can be defined as curves from a given point $p \in \overline{\mathcal{M}}$, (i.e., $\bar{\gamma}(0) = p$), with initial *velocity*, $\dot{\bar{\gamma}}(0) = \bar{\xi} \in T_{\bar{p}}\overline{\mathcal{M}}$, such that their *acceleration* is zero (a generalization of Euclidean straight lines). This characterization provides us with the following equation in local coordinates for the geodesic:

$$\frac{d^2\gamma^{\lambda}}{dt^2} + \Gamma^{\lambda}_{\mu\nu}\frac{d\gamma^{\mu}}{dt}\frac{d\gamma^{\nu}}{dt} = 0$$

where $\Gamma^{\lambda}_{\mu\nu}$ are the Christoffel symbols, $\Gamma^{\lambda}_{\mu\nu} = \frac{1}{2}g^{\lambda\sigma}\left(\frac{\partial g_{\sigma\mu}}{\partial x^{\nu}} + \frac{\partial g_{\sigma\nu}}{\partial x^{\mu}} - \frac{\partial g_{\mu\nu}}{\partial x^{\sigma}}\right)$. Additionally, the geodesics can also be derived as the curves that are minima of the energy functional

$$S(\gamma) = \int_a^b g_{\gamma(t)}(\dot{\gamma}(t), \dot{\gamma}(t))dt$$

This second perspective will prove useful for the geodesics of the attention layers.

If the initial velocity tangent vector, $\xi$, is horizontal then, $\forall t, \dot{\bar{\gamma}}(t) \in H_{\bar{\gamma}(t)}$, that is, if the velocity vector starts out as horizontal, then it will stay horizontal. We call these geodesics, *horizontal geodesics*. The curve $\gamma = \pi \circ \bar{\gamma}$ is a geodesic of the quotient manifold $\mathcal{M}$, with the same length as $\bar{\gamma}$. This also holds the other way, i.e., a geodesic in the quotient manifold can be lifted to a horizontal geodesic in the total space.

## C. Geodesic sharpness: practical concerns

### C.1. Transformers

Transformers, introduced by Vaswani et al. (2017), consist of multiheaded self-attention and feedforward layers, both wrapped in residual connections and layer normalizations. Visual transformers, in addition, tend to have convolutional layers.

Mathematically, focusing for the moment on the multi-headed attention blocks,

$$\text{MultiHead}(Q, K, V) = \begin{bmatrix} \text{head}_1, \ldots, \text{head}_h \end{bmatrix}W^o$$
$$\text{where} \quad \text{head}_i = \text{Attention}\left(QW_i^Q, KW_i^K, VW_i^V\right)$$

where $\text{Attention}(Q, K, V) = \text{softmax}\left(\frac{QK^T}{\sqrt{d_k}}\right) V$. From this we can ascertain the following symmetries:

$$1)\quad (W_i^Q, W_i^K) \to \left(W_i^Q G^{-1}, W_i^K G^T\right), \forall G \in \text{GL}_n(d_{\text{head}})$$
$$2)\quad (W_i^V, W_i^o) \to \left(W_i^V G^{-1}, W_i^o G^T\right), \forall G \in \text{GL}_n(d_{\text{head}})$$

where $W_i^o$ are the columns of $W^o$ that are relevant for the matrix multiplication with each $W_i^V$, taking into consideration the head concatenation procedure.

In the full transformer model when solving for geodesic sharpness, for each layer, we apply Eq. 5 to each $(W_i^Q, W_i^K)$ and $(W_i^V, W_i^o)$, using Eq. 11. This results in horizontal vectors $(\bar{\xi}_i^Q, \bar{\xi}_i^K)$ and $(\bar{\xi}_i^V, \bar{\xi}_i^o)$. For the non-attention parameters, $\boldsymbol{w}$, (belonging to fully connected layers, convolutional layers and layer norm), we keep to the recipe of adaptive sharpness, so that $||\bar{\xi}_{\boldsymbol{w}}|| = ||\left(\bar{\xi}_{\boldsymbol{w}} \odot |\boldsymbol{w}|^{-1}\right)||_2$. The norm of the full update vector, $\bar{\xi} = \text{concat}(\bar{\xi}_i^Q, \bar{\xi}_i^K, \bar{\xi}_i^V, \bar{\xi}_i^o, \bar{\xi}_{\boldsymbol{w}})$, where a sum over all parameters of the network is implicit, is $||\bar{\xi}||^2 = \sum\left(||(\bar{\xi}_i^Q, \bar{\xi}_i^K)||^2 + ||(\bar{\xi}_i^V, \bar{\xi}_i^o)||^2 + ||\bar{\xi}_{\boldsymbol{w}}||^2\right)$.

## C.2. Algorithm

Following the lead of Andriushchenko et al. (2023), we use Auto-PDG, proposed in Croce & Hein (2020), but now optimizing the horizontal vector $\bar{\xi}$ instead of the input. In Algorithm 1, $\ell$ is the loss over the batch we are optimizing over, S is the feasible set of horizontal vectors, $\bar{\xi}$, with norm smaller than $\rho$, and $P_S$ is the projection onto this set. $\Gamma$ are the Christoffel symbols for the parameters. $\eta$ and $W$ are fixed hyperparameters, which we keep as in Andriushchenko et al. (2023), and the two conditions in Line 20 can be found in Croce & Hein (2020). The only differences to the algorithm employed to calculate adaptive sharpness are in lines 3, 8, 10, and 12. For the metric $\langle \cdot, \cdot \rangle^{\text{mix}}$ the only differences are in the Christoffel symbols and in the Riemannian gradient ($\nabla_{\boldsymbol{G}}\ell \to \nabla_{\boldsymbol{G}}\ell\left(\boldsymbol{H}^T\boldsymbol{H}\right)^{-1}$)

---

**Algorithm 1** Auto-PGD

1: **Input:** objective function $\ell$, perturbation set $S$, $\bar{\xi}^{(0)}$, initial weights $\boldsymbol{w}^{(0)}$, $\eta$, $N_{\text{iter}}$, $W = \{w_0, \ldots, w_n\}$
2: **Output:** $\bar{\xi}_{\max}$, $\ell_{\max}$
3: $\boldsymbol{v}^{(1)} \leftarrow \boldsymbol{w}^{(0)} + \bar{\xi}^{(0)} - \frac{1}{2}\Gamma\bar{\xi}^{(0)}\bar{\xi}^{(0)}$          ▷ Perturb weights according to Eq. 5
4: $\bar{\xi}^{(1)} \leftarrow P_S\left(\bar{\xi}^{(0)} + \eta\nabla_{\bar{\xi}}\ell(\boldsymbol{v}^{(1)})\right)$
5: $\ell_{\max} \leftarrow \max\{\ell(\boldsymbol{w}^{(0)}), \ell(\boldsymbol{v}^{(1)})\}$
6: $\bar{\xi}_{\max} \leftarrow \bar{\xi}^{(0)}$ **if** $\ell_{\max} \equiv \ell(\boldsymbol{w}^{(0)})$ **else** $\bar{\xi}_{\max} \leftarrow \bar{\xi}^{(1)}$
7: **for** $k = 1$ **to** $N_{\text{iter}} - 1$ **do**
8:      $\boldsymbol{v}^{(k+1)} \leftarrow \boldsymbol{w}^{(0)} + \bar{\xi}^{(k)} - \frac{1}{2}\Gamma\bar{\xi}^{(k)}\bar{\xi}^{(k)}$          ▷ Perturb weights according to Eq. 5
9:      **if** $\boldsymbol{w}^{(0)}$ is an attention weight **then**
10:          $g \leftarrow \nabla_{\bar{\xi}}\ell(\boldsymbol{v}^{(k+1)})\boldsymbol{w}^{(0),T}\boldsymbol{w}^{(0)}$          ▷ Make attention gradients Riemannian
11:      **else**
12:          $g \leftarrow \nabla_{\bar{\xi}}\ell(\boldsymbol{v}^{(k+1)}) \odot (\boldsymbol{w}^{(0)})^2$          ▷ Make the other gradients Riemannian
13:      **end if**
14:      $\boldsymbol{z}^{(k+1)} \leftarrow P_S\left(\bar{\xi}^{(k)} + \eta g\right)$
15:      $\bar{\xi}^{(k+1)} \leftarrow P_S\left(\bar{\xi}^{(k)} + \alpha(\boldsymbol{z}^{(k+1)} - \bar{\xi}^{(k)}) + (1 - \alpha)(\bar{\xi}^{(k)} - \bar{\xi}^{(k-1)})\right)$
16:      **if** $\ell(\boldsymbol{v}^{(k+1)}) > \ell_{\max}$ **then**
17:          $\bar{\xi}_{\max} \leftarrow \bar{\xi}^{(k+1)}$ and $\ell_{\max} \leftarrow \ell(\boldsymbol{v}^{(k+1)})$
18:      **end if**
19:      **if** $k \in W$ **then**
20:          **if** Condition 1 **or** Condition 2 **then**
21:              $\eta \leftarrow \eta/2$ and $\boldsymbol{w}^{(k+1)} \leftarrow \boldsymbol{w}_{\max}$
22:          **end if**
23:      **end if**
24: **end for**

---

### C.3. Complexity

Geodesic sharpness is slightly more expensive than adaptive sharpness in the following sense: Our approach consists of three steps: 1) perturbing the weights according to Eq. 5, 2) optimizing the perturbations with gradient descent, and 3) projecting them onto the feasible set, i.e. horizontal vectors within the geodesic ball with a small enough norm.

Steps 1) and 2) are also present in adaptive sharpness. Step 1) in our approach is slightly more expensive because we need to evaluate the quadratic form that involves the Christoffel symbols (given by Eq. 11 and Eq. 12); this step introduces $n_{\text{params}}$ weight matrix multiplications, but these are quite efficient. Making the gradients Riemannian, costs another $n_{\text{params}}$ weight matrix multiplications. Neither of these bottleneck our approach. For $\langle \cdot, \cdot \rangle^{\text{inv}}$, Step 3) requires solving a Sylvester equation to project the direction of the updated geodesic back onto the horizontal space. This solve is cubic in $h$ (Kirrinnis, 2001), but $h$ is usually small (e.g. $h = 64$ in the ImageNet and BERT experiments). For $\langle \cdot, \cdot \rangle^{\text{mix}}$, only efficient matrix multiplications are required.

On practical transformers, we expect the bottleneck to be the forward and backward propagations, just like in adaptive sharpness.

## D. Geodesic sharpness: Scalar Toy model

To make our approach explicit, we illustrate it on a NN with two scalar parameters $G$ and $H$, square loss, and a single (scalar) training point $(x, y)$. We use $\langle \cdot, \cdot \rangle^{\text{inv}}$ throughout. For this example, everything is analytically tractable. We also contrast our sharpness measure with previously proposed ones to highlight its invariance.

Since we require full column-rank, our parameter space is $\mathcal{M} = \mathbb{R}_* \times \mathbb{R}_*$ with $\mathbb{R}_* = \mathbb{R} \setminus \{0\}$.

**Metric:** At a point $(G, H) \in \mathcal{M}$, for two tangent vectors $\boldsymbol{\eta} = (\eta_G, \eta_H), \boldsymbol{\nu} = (\nu_G, \nu_H) \in T_{(G,H)}\mathcal{M}$, we have

$$\langle \boldsymbol{\eta}, \boldsymbol{\nu} \rangle^{\text{inv}} = \frac{\eta_G \nu_G}{G^2} + \frac{\eta_H \nu_H}{H^2} = \eta^\top \underbrace{\begin{pmatrix} \frac{1}{G^2} & 0 \\ 0 & \frac{1}{H^2} \end{pmatrix}}_{g_{kl}} \nu \tag{18}$$

We denote the inverse metric by $g^{kl} = \begin{pmatrix} G^2 & 0 \\ 0 & H^2 \end{pmatrix}$

**Horizontal space:** $H_{(G,H)} = \{(\eta_G, \eta_H) \in T_{(G,H)}\mathcal{M} \mid \frac{\eta_G}{G} = \frac{\eta_H}{H}\}$

**Geodesics:** To compute the geodesics on the quotient space, we need the Christoffel symbols $\Gamma^i_{km}$.

Using a coordinate system $(p^1, p^2) = (G, H)$, we have the following equation for a geodesic $\gamma(t) = (\gamma_G(t), \gamma_H(t))$, with initial conditions $\gamma(0) = (G_0, H_0)$ and $\dot{\gamma}(0) = (\eta_{G_0}, \eta_{H_0})$

$$\frac{d^2 \gamma_G}{dt^2} + \Gamma^1_{11} \left( \frac{d\gamma_G}{dt} \right)^2 = 0$$

and similarly for $H$ with $\Gamma^2_{22}$ instead of $\Gamma^1_{11}$.

The Christoffel symbols can be found using the metric, $g$, and its inverse. Using the Einstein notation and denoting the inverse of $g$ by the use of upper indices:

$$\Gamma^i_{kl} = \frac{1}{2} g^{im} \left( \frac{\partial g_{mk}}{\partial x^l} + \frac{\partial g_{ml}}{\partial x^k} - \frac{\partial g_{kl}}{\partial x^m} \right)$$

Then

$$\Gamma^1_{11} = \frac{1}{2} g^{1m} \left( \frac{\partial g_{m1}}{\partial p^1} + \frac{\partial g_{m1}}{\partial p^1} - \frac{\partial g_{kl}}{\partial p^m} \right) = -\frac{1}{G}$$

$$\Gamma^2_{22} = -\frac{1}{H}$$

All other Christoffel symbols are 0. Our geodesic equations then become (we omit the derivation for H, which is identical but with $G \leftrightarrow H$)

$$\frac{d^2\gamma_G}{dt^2} - \frac{1}{\gamma_G}\left(\frac{d\gamma_G}{dt}\right)^2 = 0$$

This ODE has the (unique) solution $\gamma_G(t) = A_G \exp(b_G t)$. Taking into account the initial conditions, $A_G = G_0$, $A_H = H_0$ and due to the definition of the horizontal space, $b_G = \frac{\eta_G}{G_0}$ and $b_H = \frac{\eta_H}{H_0}$, this becomes

$$\gamma(t) = \left(G_0 \exp\left(\frac{\eta_G}{G_0}t\right), H_0 \exp\left(\frac{\eta_H}{H_0}t\right)\right)$$

One important detail to note is that these geodesics are not complete, that is, not all two points can be connected by a geodesic. Points with different signs cannot be connected, which makes sense since we excluded the origin from the acceptable parameters and in 1D we need to cross it to connect points with differing signs. All points that lie in the same quadrant as $(G_0, H_0)$ can be connected through a geodesic.

Putting it all together

$$S_{\max}^\rho((G_0, H_0)) = \left[\max_{||b||\leq\rho} x^2 G_0^2 H_0^2(\exp(4b) - 1) - 2yxG_0H_0(\exp(2b) - 1)\right], \tag{19}$$

Letting $y_0 = G_0 H_0 x$, this becomes:

$$S_{\max}^\rho((G_0, H_0)) = \left[\max_{||b||\leq\rho} y_0^2(\exp(4b) - 1) - 2yy_0(\exp(2b) - 1)\right], \tag{20}$$

Since $\eta_H$ is completely determined by $\eta_G$ we can ignore the maximization over it.

Since in practice we'll take $\rho \ll 1$, we Taylor expand to get

$$S_{\max}^\rho \approx 4\rho|y_0||y - y_0|$$

This presents an issue when the residual, $|y - y_0|$, is zero, so we can also expand to second order, to get, when $|y - y_0| \approx 0$

$$S_{\max}^\rho \propto \rho^2|y_0||y - 2y_0| = 2\rho^2 y_0^2$$

This is, up to constants, just $||G \odot H||_2^2$. This is also invariant to $GL_1$ transformations, as expected.

Very close to the minimum we only capture (second-order in $\rho$) properties of the network, a bit further away from it we capture a (first-order in $\rho$) mix of data and network properties.

**Comparison with more traditional measures:** The local average and worst case Euclidean sharpness (at a minimum) are

$$S_{\text{avg}} = \text{Tr}\,\nabla^2 L_S = G^2 + H^2$$
$$S_{\max} = \lambda_{\max}(\nabla^2 L_S) = G^2 + H^2$$

Adaptive sharpness is defined as

$$S_{\text{avg}}^\rho(w, c) = \mathbb{E}_{S\sim\mathbb{P}_m}\left[L_S(w + \delta) - L_S(w)\right], \quad \delta \sim \mathcal{N}(0, \rho^2\text{diag}(c^2))$$
$$S_{\max}^\rho(w, c) = \mathbb{E}_{S\sim\mathbb{P}_m}\left[\max_{||\delta\odot c^{-1}||_p\leq\rho} L_S(w + \delta) - L_S(w)\right],$$

By picking $c$ very carefully one can get

$$S_{\text{avg}}^\rho(w, c) = |GH|$$
$$S_{\max}^\rho(w, c) = |GH|$$

By contrast, in our approach there is no need for careful hyperparameter choices

**Geodesic flatness with more data points:**   How does the geodesic flatness look like with more data points?

$$L_S(G, H) = \frac{1}{n} \sum_{i=1}^{n} (GHx_i - y_i)^2$$

which leads to (defining $y_i^0 = GHx_i$):

$$S_{\max}^{\rho} = \max_b \frac{1}{n} \sum_{i=1}^{n} \left[ (y_i^0)^2 \left( \exp\left(\frac{b}{|b|} 2\sqrt{2}\rho\right) - 1 \right) - 2yy_i^0 \left( \exp\left(\frac{b}{|b|}\sqrt{2}\rho\right) - 1 \right) \right] \tag{21}$$

Taylor expanding (in $\rho$) once more, we see that

$$S_{\max}^{\rho} \approx \max_b \frac{1}{n} \sum_{i=1}^{n} \left[ 2\sqrt{2}\rho \frac{b}{|b|} y_i^0 (y_i^0 - y) + 2\rho^2 (y_i^0)^2 \right] \tag{22}$$

Which $b$ maximizes Eq. 22, depends on the sign of $\sum_{i=1}^{n} \left[ y_i^0 (y_i^0 - y) \right]$: $b < 0$ if the sum is negative, the reverse if the opposite is true.

### D.1. Traditional flatness

In Figure 8 we extend Figure 1 to include the trace of the Hessian, both Euclidean and Riemannian. The trace of the network Hessian is a quantity that can be used to quantify flatness. We plot, for the scalar toy model, the level sets of: a) the loss function; b) the Euclidean and Riemannian gradient; c) the traces of the Euclidean and Riemannian network Hessian. Several features of the plots are important to note: a) the Riemannian version of the gradient and Hessian have the same level set geometry as the loss function; b) both the Riemannian gradient norm and the trace of the Riemannian Hessian have smaller values throughout than their Euclidean equivalents; c) the trace of the Riemannian Hessian actually reaches 0 when at the local minimum, whereas the Euclidean Hessian actually attains its highest value there; d) the Euclidean trace of the Hessian cannot distinguish between a minimum and a maximum whereas the Riemannian trace can actually do so. Even for simple flatness measures, correcting for the quotient geometry can provide a much clearer picture.

## E. Geodesic sharpness: Diagonal networks in full generality

### E.1. Metric (6)

**Metric:**   At a point $(\boldsymbol{u}, \boldsymbol{v}) \in \mathcal{M}$, for two tangent vectors $\boldsymbol{\eta} = (\boldsymbol{\eta_u}, \boldsymbol{\eta_v})$, $\boldsymbol{\nu} = (\boldsymbol{\nu_u}, \boldsymbol{\nu_v}) \in T_{(\boldsymbol{u}, \boldsymbol{v})}\mathcal{M}$, we have

$$\langle \boldsymbol{\eta}, \boldsymbol{\nu} \rangle^{\text{inv}} = \sum_{i=1}^{d} \frac{\boldsymbol{\eta_u^i} \boldsymbol{\nu_u^i}}{(\boldsymbol{u^i})^2} + \frac{\boldsymbol{\eta_v^i} \boldsymbol{\nu_v^i}}{(\boldsymbol{v^i})^2} \tag{23}$$

**Horizontal space:**   $H_{(\boldsymbol{u}, \boldsymbol{v})} = \{(\eta_u, \eta_v) \in T_{(\boldsymbol{u}, \boldsymbol{v})}\mathcal{M} \mid \frac{\eta_u^i}{\boldsymbol{u^i}} = \frac{\eta_v^i}{\boldsymbol{v^i}} \quad \forall i \in \{1, \ldots, d\}\}$

**Geodesics:**   We define $\boldsymbol{b^i} = \frac{\eta_u^i}{\boldsymbol{u^i}} = \frac{\eta_v^i}{\boldsymbol{v^i}} \forall i \in \{1, \ldots, d\}$, so that

$$\boldsymbol{\gamma}(t)^i = (\boldsymbol{u}(t), \boldsymbol{v}(t)) = \left( \boldsymbol{u_0^i} \exp(\boldsymbol{b_i} t), \boldsymbol{v_0^i} \exp(\boldsymbol{b_i} t) \right) \forall i \in \{1, \ldots, d\} \tag{24}$$

where $\boldsymbol{u_0^i}$ and $\boldsymbol{v_0^i}$ are the initial positions for our parameters, i.e., the parameters that the network actually learned.

**Geodesic sharpness:**   We assume that in what follows $\boldsymbol{X}^T \boldsymbol{X} = Id_d$, and we denote $\boldsymbol{\beta_0} = \boldsymbol{u_0} \odot \boldsymbol{v_0}$, $\boldsymbol{\gamma_t} = \left( \exp(2\boldsymbol{b^1} t), \ldots \exp(2\boldsymbol{B^d} t) \right)$, $\boldsymbol{\beta_t} = (\boldsymbol{u_t} \odot \boldsymbol{v_t}) = \boldsymbol{\beta_0} \odot \boldsymbol{\gamma_t}$, $\boldsymbol{\beta_*} = \boldsymbol{X}^T y$. Note that $\boldsymbol{\beta_*}$ is just the optimal least squares predictor when $\boldsymbol{X}^T \boldsymbol{X} = Id$. With this notation

$$S_{\max} = \max_{||\boldsymbol{b}|| \leq \rho} \sum_{i}^{d} \left[ (\boldsymbol{\beta_0^i})^2 (\boldsymbol{\gamma_t} \odot \boldsymbol{\gamma_t} - 1) \right] - 2(\boldsymbol{\beta_0} \odot \boldsymbol{\gamma_t} - 1)^T \boldsymbol{\beta_*} \tag{25}$$

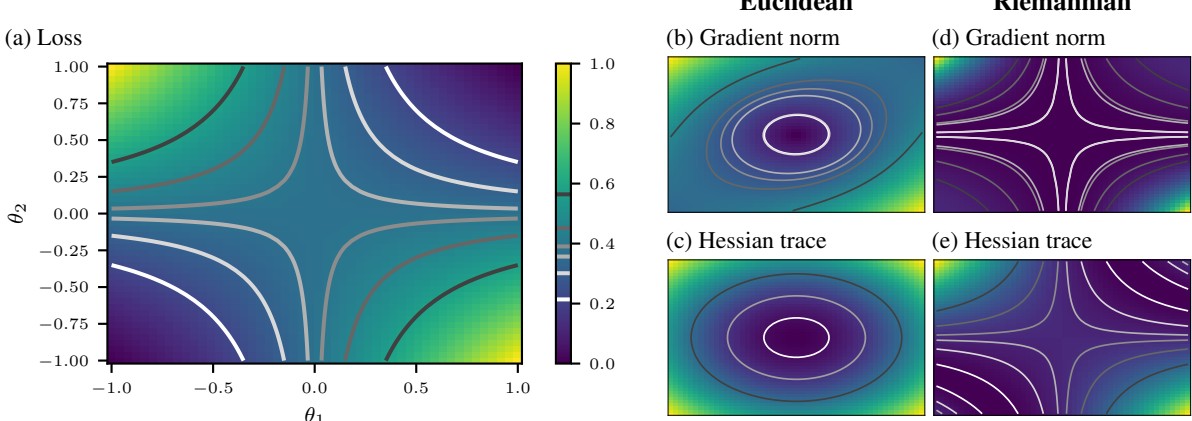

Figure 8: **Quantities from the Riemannian quotient manifold respect the loss landscape's symmetry; Euclidean quantities do not.** We use a synthetic least squares regression task with a two-layer NN $x \mapsto \theta_2 \theta_1 x$ with scalar parameters $\theta_i \in \mathbb{R}$ and input $x \in \mathbb{R}$. The NN is re-scale invariant, i.e. has GL(1) symmetry: For any $\alpha \in \mathbb{R} \setminus \{0\}$, the parameters $(\theta_1', \theta_2') = (\alpha^{-1}\theta_1, \alpha\theta_2)$ represent the same function. (a) The loss function inherits this symmetry and has hyperbolic level sets. (b,c) The Euclidean gradient norm does not share the loss function's geometry and changes throughout an orbit where the NN function remains constant. (d,e) The Riemannian gradient norm and Hessian trace follow the loss function's symmetry and remain constant throughout an orbit, i.e. they do not suffer from ambiguities for two points in parameter space that represent the same NN function. All quantities were normalized to $[0; 1]$ and we fixed six points in parameter space and computed the level sets running through them to illustrate the geometry.

At a first glance, this expression does not seem to have a simple interpretation, but we Taylor expand it to second order in $\boldsymbol{B}$ (since $\rho$ is supposed to be small):

$$S_{\text{max}} \approx \max_{||\boldsymbol{b}|| \leq \rho} 4\boldsymbol{b}^T \boldsymbol{r} + 4\boldsymbol{b}^T \boldsymbol{D}_{\boldsymbol{\beta}_0, \boldsymbol{\beta}_*} \boldsymbol{b} \tag{26}$$

where $\boldsymbol{r} = \{\boldsymbol{\beta}_0^i(\boldsymbol{\beta}_0^i - \boldsymbol{\beta}_*^i), i = 1, \ldots, d\}$, $\boldsymbol{r}' = \{(\boldsymbol{\beta}_0^i - \boldsymbol{\beta}_*^i), i = 1, \ldots, d\}$ and $\boldsymbol{D}_{\boldsymbol{\beta}_0, \boldsymbol{\beta}_*} = diag(\boldsymbol{\beta}_0^i(2\boldsymbol{\beta}_0^i - \boldsymbol{\beta}_*^i)) = diag(\boldsymbol{\beta}_0^i(\boldsymbol{\beta}_0^i + (\boldsymbol{r}')^i))$. We separate the analysis of Eq.26 into three cases:

**case a): $\boldsymbol{r} \neq 0$ and first order suffices**   Eq.26 becomes

$$S_{\text{max}} = \max_{||\boldsymbol{b}|| \leq \rho} 4\boldsymbol{b}^T \boldsymbol{r}$$

with solution $S_{\text{max}} = 4\rho||\boldsymbol{r}||$. This is essentially the gradient norm– a useful quantity for understanding generalization (Zhao et al., 2022b).

**case b): $\boldsymbol{r} = 0$**   Here we necessarily have to consider the second order terms, so that Eq.26 becomes

$$S_{\text{max}} = \max_{||\boldsymbol{b}|| \leq \rho} 4\boldsymbol{b}^T \boldsymbol{D}_{\boldsymbol{\beta}_0, \boldsymbol{\beta}_*} \boldsymbol{b}$$

This has the well known solution of $S_{\text{max}} = \rho^2 \lambda_{\text{max}}(\boldsymbol{D}_{\boldsymbol{\beta}_0, \boldsymbol{\beta}_*}) = \rho^2 \max((\beta_0^i)^2)$. This is just $||\boldsymbol{\beta}||_\infty^2$, which is the square of what we would get by using adaptive sharpness, Eq.39, with a very carefully chosen hyper-parameter $\boldsymbol{c}$. This is a quantity that is useful when our ground-truth, $\boldsymbol{\beta}^*$ is dense.

**case c): $\boldsymbol{r} \neq 0$ and we need both first and second order terms**   In this case, Eq.26 needs to be considered in full, and we solve the maximization problem using Lagrange multipliers. The Lagrangian will be

$$\mathcal{L} = -4\boldsymbol{b}^T \boldsymbol{r} - 4\boldsymbol{b}^T \boldsymbol{D}_{\boldsymbol{\beta}_0, \boldsymbol{\beta}_*} \boldsymbol{b} + \lambda(\boldsymbol{b}^T \boldsymbol{b} - \rho^2)$$

The KKT conditions then are

$$\frac{\partial \mathcal{L}}{\partial \boldsymbol{b}} = -4\boldsymbol{r} - 8\boldsymbol{D}_{\boldsymbol{\beta}_0, \boldsymbol{\beta}_*} \boldsymbol{b} + 2\lambda \boldsymbol{b} = 0 \tag{27}$$

$$\lambda(\boldsymbol{b}^T \boldsymbol{B} - \rho^2) = 0 \tag{28}$$

$$\lambda \geq 0 \tag{29}$$

If the constraint is not active, then $\lambda = 0$ and

$$\boldsymbol{b}_* = -\frac{1}{2} \boldsymbol{D}_{\boldsymbol{\beta}_0, \boldsymbol{\beta}_*}^{-1} \boldsymbol{r}$$

In practice, unless $\rho$ is large the constraint will always be active, in which case 27 becomes

$$-4\boldsymbol{r} - 8\boldsymbol{D}_{\boldsymbol{\beta}_0, \boldsymbol{\beta}_*} \boldsymbol{b} + 2\lambda(\boldsymbol{B}) = 0$$

$$(\boldsymbol{b}^T \boldsymbol{b} - \rho^2) = 0$$

$$\lambda \geq 0$$

this then becomes

$$\boldsymbol{B}_* = 2 \left( \lambda I - 4\boldsymbol{D}_{\boldsymbol{\beta}_0, \boldsymbol{\beta}_*} \right)^{-1} \boldsymbol{r}$$

$$4 \sum_i^d \frac{(\boldsymbol{r}^i)^2}{\left( \lambda - 4(\boldsymbol{\beta}_0^i(\boldsymbol{\beta}_0^i + \boldsymbol{r}')) \right)^2} = \rho^2$$

$$\lambda \geq 0$$

### E.2. Metric (7)

We follow the same approach as in the previous section. The main difference will be in the form of the geodesics: $\boldsymbol{u}(t) \odot \boldsymbol{v}(t) = (\boldsymbol{u}_0 \odot \boldsymbol{v}_0) \odot (1 + 2\boldsymbol{b}t)$, where $\boldsymbol{b}^i = \frac{\eta_{\boldsymbol{u}}^i}{\boldsymbol{u}^i} = \frac{\eta_{\boldsymbol{v}}^i}{\boldsymbol{v}^i}$, as in the previous section. This essentially treats the two-layer neural network as if it were a single layer, with predictor $\beta = \boldsymbol{u} \odot \boldsymbol{v}$, that it then perturbs linearly to determine sharpness. For $\langle \cdot, \cdot \rangle^{\text{mix}}$, and denoting by $\boldsymbol{D}_{\boldsymbol{\beta}} = diag(\boldsymbol{\beta}_0^i)$:

$$S_{max} = \max_{||\eta||^{\text{mix}} \leq \rho} 4 \left[ \boldsymbol{b}^T (\boldsymbol{\beta}_0 - \boldsymbol{\beta}_*) + \boldsymbol{b}^T \boldsymbol{D}_{\boldsymbol{\beta}}^2 \boldsymbol{b} \right] \tag{30}$$

We also have that

$$(||\eta||^{\text{mix}})^2 = \left[ \ldots + (\boldsymbol{v}^i)^2 (\eta_{\boldsymbol{u}}^i)^2 + (\boldsymbol{u}^i)^2 (\eta_{\boldsymbol{v}}^i)^2 + \ldots \right] \tag{31}$$

$$= \left[ \ldots + (\boldsymbol{v}^i)^2 (\boldsymbol{u}^i)^2 \left( \frac{(\eta_{\boldsymbol{u}}^i)^2}{(\boldsymbol{u}^i)^2} + \frac{(\eta_{\boldsymbol{v}}^i)^2}{(\boldsymbol{v}^i)^2} \right) + \ldots \right] \tag{32}$$

$$= \left[ \ldots + 2(\boldsymbol{v}^i)^2 (\boldsymbol{u}^i)^2 (\boldsymbol{b}^i)^2 + \ldots \right] = ||2\boldsymbol{D}_{\boldsymbol{\beta}_0} \boldsymbol{b}||_2 \tag{33}$$

Substituting $2\boldsymbol{D}_{\boldsymbol{\beta}_0} \boldsymbol{b} = \boldsymbol{\delta}$, Eq. 30 becomes

$$S_{max} = \max_{||\boldsymbol{\delta}|| \leq \rho} \left[ \boldsymbol{\delta}^T (\boldsymbol{\beta}_0 - \boldsymbol{\beta}_*) + \boldsymbol{\delta}^T \boldsymbol{\delta} \right] \tag{34}$$

with the solution (up to constants)

$$S_{max} = \rho ||\boldsymbol{\beta}_0 - \boldsymbol{\beta}_*||_2 \tag{35}$$

## F. Geodesic Sharpness: $GL_1$ symmetry and Adaptive Sharpness

What happens if instead of a general $GL_n$ symmetry, we factor out a $GL_1$ re-scaling symmetry? That is, we identify, element-wise, $(\bar{x}, \bar{y}) \sim (\bar{x}'\bar{y}')$ if $\exists \alpha \in \mathbb{R}_* = \mathbb{R} \setminus \{0\}$ s.t. $\bar{x} = \alpha\bar{x}'$ and $\bar{y} = \alpha^{-1}\bar{y}'$.

This is the symmetry present in diagonal networks, and so we utilize the metric given by Eq. 6, reproduced below for convenience of the reader:

$$g\left[(\eta_{\boldsymbol{u}}, \eta_{\boldsymbol{v}}), (\nu_{\boldsymbol{u}}, \nu_{\boldsymbol{v}})\right] = \sum_{i=1}^{d} \frac{\eta_{\boldsymbol{u}}^i \nu_{\boldsymbol{u}}^i}{(\boldsymbol{u}^i)^2} + \frac{\eta_{\boldsymbol{v}}^i \nu_{\boldsymbol{v}}^i}{(\boldsymbol{v}^i)^2} \tag{36}$$

Note that this metric is equivalent to the following metric:

$$g\left[(\eta_{\boldsymbol{u}}, \eta_{\boldsymbol{v}}), (\nu_{\boldsymbol{u}}, \nu_{\boldsymbol{v}})\right] = g\left[(\eta_{\boldsymbol{u}}/|\boldsymbol{u}|, \eta_{\boldsymbol{v}}/|\boldsymbol{v}|), (\nu_{\boldsymbol{u}}/|\boldsymbol{u}|, \nu_{\boldsymbol{v}}/|\boldsymbol{v}|)\right]_{\text{euc}} \tag{37}$$

where $g_{\text{euc}}$ is the usual Euclidean metric and the division is taken to be element-wise. Denoting the concatenation of all tangent vectors by $\xi$, the concatenation of all parameters by $\boldsymbol{w}$, we have $||\xi|| = ||\xi/|\boldsymbol{w}|||_2$.

In this situation Eq. 4 becomes ($\gamma$ denotes our geodesics as usual)

$$S_{\max}^\rho(w, c) = \mathbb{E}_{\mathbb{S} \sim \mathbb{D}} \left[ \max_{||\xi/|\boldsymbol{w}|||_2 \leq \rho} L_S(\bar{\gamma}_{\bar{\xi}}(1)) - L_S(\bar{\gamma}_{\bar{\xi}}(0)) \right], \tag{38}$$

If we then ignore the corrections induced by the geometry of the metric on the geodesics, i.e., take $\bar{\gamma}_{\bar{\xi}}(1) = \bar{\gamma}_{\bar{\xi}}(0) + \bar{\xi} = \boldsymbol{w} + \bar{\xi}$, then we get

$$S_{\max}^\rho(w, c) = \mathbb{E}_{\mathbb{S} \sim \mathbb{D}} \left[ \max_{||\xi/|\boldsymbol{w}|||_2 \leq \rho} L_S(\boldsymbol{w} + \xi) - L_S(\boldsymbol{w}) \right] \tag{39}$$

which is exactly the formula for adaptive sharpness.

## G. Geodesic Sharpness: Ablations

In this appendix we conduct ablation studies on geodesic sharpness (Equation (4)). There are two main components to our recipe that differ from adaptive sharpness: a) the norm $||\bar{\xi}||$; b) the weight update formula, which instead of the usual $\boldsymbol{w}^i = \boldsymbol{w}^i + \bar{\xi}$ takes into account the curvature induced by the parameter space symmetries $\boldsymbol{w}^i = \boldsymbol{w}^i + \bar{\xi}^i - \frac{1}{2}\Gamma_{kl}^i \bar{\xi}^k \bar{\xi}^l$. Below we turn off these components one by one and re-compute the resulting sharpness on MNLI using the BERT models described in Section 5.3.2.

**Metric (9):** In Figure 9 we show the results for our ablation studies using metric (9). The norm component is much more impactful than the second-order weight corrections. Turning off the second-order weight corrections results in a small performance drop only.

**Metric (10):** In Figure 10 we show the results for our ablation studies using metric (10). The norm component is still much more impactful than the second-order weight corrections for this metric, but now the second-order weight corrections are essential, and without them sharpness loses a considerable amount of predictive power.

## H. Geodesic Sharpness: Ranks and Relaxation

### H.1. Ranks: how natural is Assumption 5.1?

In general, in non-linear networks there is a tendency towards low-rank representations, which might make Assumption 5.1 seem excessive and counter to realistic situations. However, while the learned $W_Q W_K^T$ tend to be low-rank, $W_Q$ and $W_K$ (on which Assumption 5.1 ought to apply) themselves are usually high/full (column) rank (Yu & Wu, 2023).

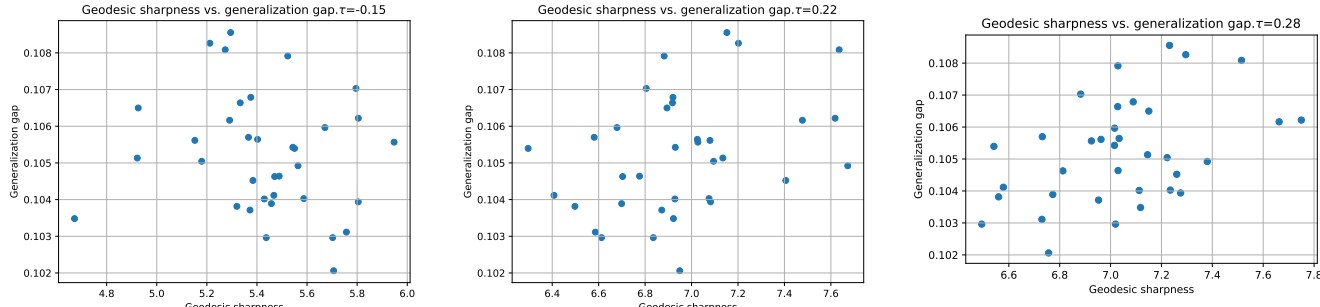

Figure 9: The generalization gap on the MNLI dev matched set (Williams et al., 2018) vs. worst-case adaptive sharpness with metric (9) is shown for 35 models from (McCoy et al., 2020). On the left we plot the results when we turn off the corrected norm, and on the middle when we turn off the second-order weight corrections. Right are the results with no ablations.

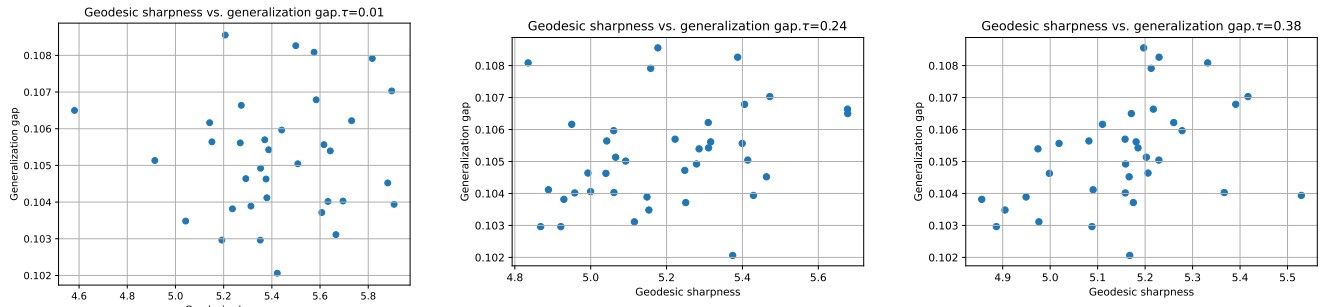

Figure 10: The generalization gap on the MNLI dev matched set (Williams et al., 2018) vs. worst-case adaptive sharpness with metric (10) is shown for 35 models from (McCoy et al., 2020). On the left we plot the results when we turn off the corrected norm, and on the middle when we turn off the second-order weight corrections. On the right are the results with no ablations.

## H.2. Relaxation

Due to the definition of metric 9, we need to invert matrices of the type of $W_Q^T W_Q$. When these are not full-rank, numerical stability can suffer. Due to floating-point precision rounding errors, in practice $W_Q^T W_Q$ is always invertible, but sometimes the inverted matrices have huge singular values. To combat this, we introduce a relaxation parameter, so that $W_Q^T W_Q \to W_Q^T W_Q + \epsilon I_h$, which dampens the resulting singular values. Although we cannot take it to be exactly zero, as long as it is small enough, numerical stability is improved and the results remain roughly the same. We study the effects of varying this parameter on our results empirically below (Figure 11), using the same setup as in Section 5.3.2. The results are not significantly affected by the variation of this parameter.

## I. Additional Derivations and Proofs

### I.1. Proof that Eq. 9 defines a valid Riemannian metric

Eq. 9 defines a valid metric on the total space $\overline{\mathcal{M}}$ if it is smooth, and for each point $(\bar{G}, \bar{H}) \in \overline{\mathcal{M}}$ it defines a valid inner product on the tangent space $T_{(\bar{G},\bar{H})}\overline{\mathcal{M}}$. That it is smooth is obvious, so we show that $\langle \bar{\eta}, \bar{\zeta} \rangle_{(\bar{G},\bar{H})} = \mathrm{Tr}\left( (\boldsymbol{G}^\top \boldsymbol{G})^{-1} \bar{\eta}_{\boldsymbol{G}}^\top \bar{\zeta}_{\boldsymbol{G}} + (\boldsymbol{H}^\top \boldsymbol{H})^{-1} \bar{\eta}_{\boldsymbol{H}}^\top \bar{\zeta}_{\boldsymbol{H}} \right)$ defines a valid inner product:

   (i) *Symmetry* $\langle \bar{\eta}, \bar{\zeta} \rangle = \langle \bar{\zeta}, \bar{\eta} \rangle$: omitting the $\boldsymbol{H}$ term as it is identical, $\langle \bar{\eta}, \bar{\zeta} \rangle = \mathrm{Tr}\left( (\boldsymbol{G}^\top \boldsymbol{G})^{-1} \bar{\eta}_{\boldsymbol{G}}^\top \bar{\zeta}_{\boldsymbol{G}} \right) = \mathrm{Tr}\left( (\boldsymbol{G}^\top \boldsymbol{G})^{-1} \bar{\zeta}_{\boldsymbol{G}}^\top \bar{\eta}_{\boldsymbol{G}} \right) = \langle \bar{\zeta}, \bar{\eta} \rangle$ ;

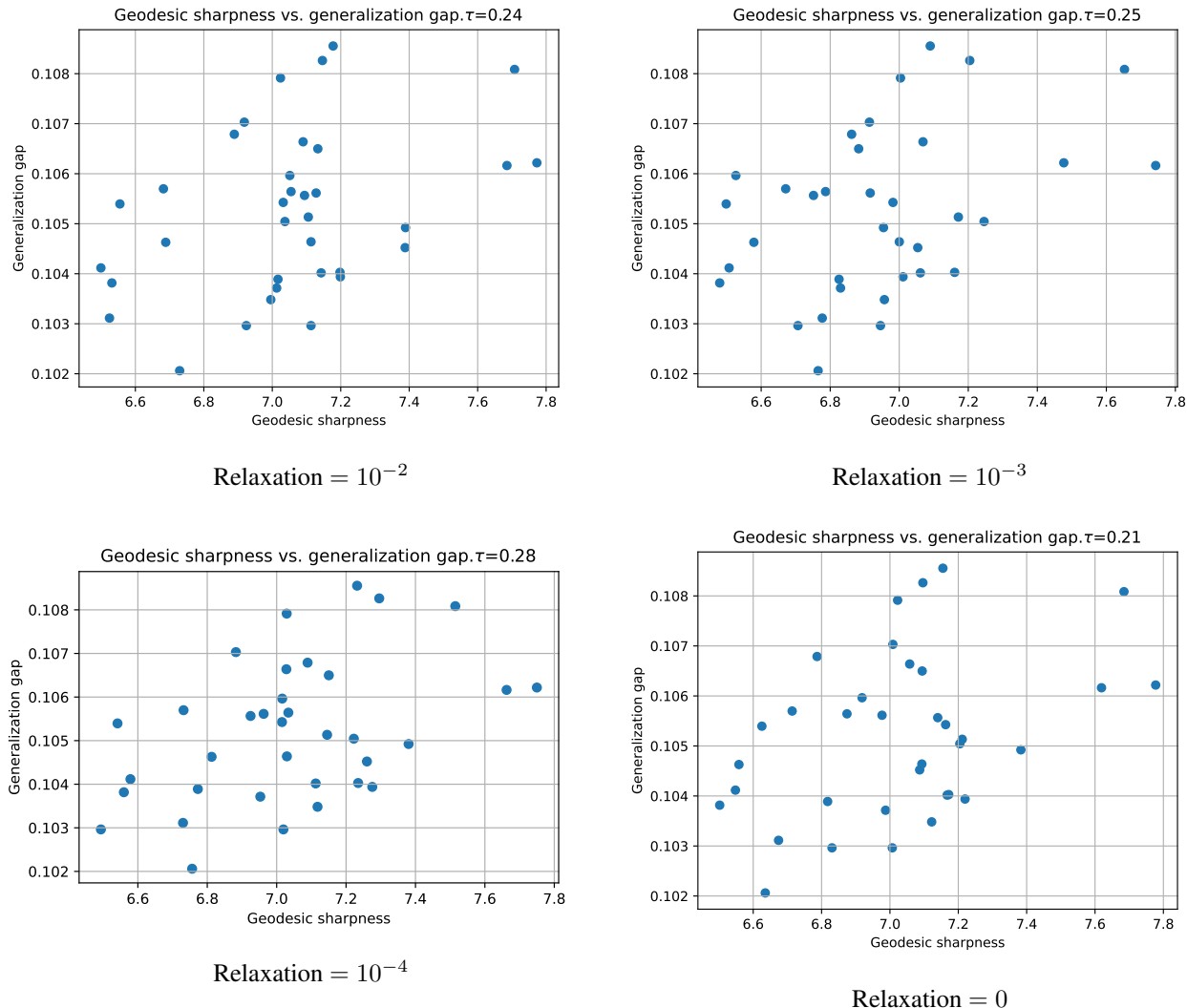

Figure 11: The generalization gap on the MNLI dev matched set (Williams et al., 2018) vs. worst-case adaptive sharpness (left) and geodesic sharpness ($\langle \cdot, \cdot \rangle^{\text{inv}}$), is shown for 35 models from (McCoy et al., 2020). Only the relaxation parameter differs between plots. The results stay broadly the same.

(ii) *Bilinearity* $\langle a\bar{\eta} + b\bar{\zeta}, \bar{\lambda} \rangle = a\langle \bar{\eta}, \bar{\lambda} \rangle + b\langle \bar{\zeta}, \bar{\lambda} \rangle = \langle \bar{\lambda}, a\bar{\eta} + b\bar{\zeta} \rangle$: follows by linearity of the trace;

(iii) *Positive Definiteness* $\langle \bar{\eta}, \bar{\eta} \rangle \geq 0$: using assumption 5.1, $\boldsymbol{G}^T \boldsymbol{G}$ is invertible and is positive-definite; this means that $(\boldsymbol{G}^T \boldsymbol{G})^{-1}$ is also positive-definite, and so $\langle \bar{\eta}, \bar{\eta} \rangle \geq 0$, with equality only when $\bar{\eta} = 0$.

The proof that Equation (10) defines a valid metric is analogous.

### I.2. Derivation of the geodesic corrections for attention

We apply the Euler-Lagrange formalism to the energy functional to derive the geodesic equation on the attention quotient manifold, and hence $\Gamma^i_{kl} \bar{\xi}^k_{\boldsymbol{G}} \bar{\xi}^l_{\boldsymbol{G}}$, remembering that geodesics, in local coordinates, obey the equation $\frac{d^2 \gamma^i}{dt^2} + \Gamma^i_{kl} \frac{d\gamma^k}{dt} \frac{d\gamma^l}{dt} = 0$.

Starting from

$$E(\gamma) = \int_0^1 \mathcal{L} \, dt = \int_0^1 \langle \dot{\gamma}(t), \dot{\gamma}(t) \rangle_{\gamma(t)} dt \tag{40}$$

$$= \int_0^1 \left[ \text{Tr}\big(\gamma_{\boldsymbol{G}}(t)^T \gamma_{\boldsymbol{G}}(t)\big) \dot{\gamma}_{\boldsymbol{G}}(t)^T \dot{\gamma}_{\boldsymbol{G}}(t) + \text{Tr}\big(\gamma_{\boldsymbol{H}}(t)^T \gamma_{\boldsymbol{H}}(t)\big) \dot{\gamma}_{\boldsymbol{H}}(t)^T \dot{\gamma}_{\boldsymbol{H}}(t) \right] dt \tag{41}$$

,

The Euler-Lagrange equation, for the $\boldsymbol{G}$ part only, reads

$$\frac{d}{dt}\left(\frac{\partial \mathcal{L}}{\partial \dot{\boldsymbol{G}}}\right) - \frac{\partial \mathcal{L}}{\partial \boldsymbol{G}} = 0 \tag{42}$$

We have

$$\frac{\partial \mathcal{L}}{\partial \boldsymbol{G}} = -2\boldsymbol{G} \left(\boldsymbol{G}^T \boldsymbol{G}\right)^{-1} \left(\dot{\boldsymbol{G}}^T \dot{\boldsymbol{G}}\right) \left(\boldsymbol{G}^T \boldsymbol{G}\right)^{-1} \tag{43}$$

$$\frac{d}{dt}\left(\frac{\partial \mathcal{L}}{\partial \dot{\boldsymbol{G}}}\right) = 2\ddot{\boldsymbol{G}} \left(\boldsymbol{G}^T \boldsymbol{G}\right)^{-1} - 2\dot{\boldsymbol{G}} \left(\boldsymbol{G}^T \boldsymbol{G}\right)^{-1} \left(\dot{\boldsymbol{G}}^T \boldsymbol{G} + \boldsymbol{G}^T \dot{\boldsymbol{G}}\right) \left(\boldsymbol{G}^T \boldsymbol{G}\right)^{-1} \tag{44}$$

So that Eq. 42 becomes:

$$\ddot{\boldsymbol{G}} - \dot{\boldsymbol{G}} \left(\boldsymbol{G}^T \boldsymbol{G}\right)^{-1} \left(\dot{\boldsymbol{G}}^T \boldsymbol{G} + \boldsymbol{G}^T \dot{\boldsymbol{G}}\right) + \boldsymbol{G} \left(\boldsymbol{G}^T \boldsymbol{G}\right)^{-1} \left(\dot{\boldsymbol{G}}^T \dot{\boldsymbol{G}}\right) = 0 \tag{45}$$

From which we read

$$\Gamma^i_{kl} \bar{\xi}^k_{\boldsymbol{G}} \bar{\xi}^l_{\boldsymbol{G}} = \left[ -\bar{\xi} \left(\boldsymbol{G}^T \boldsymbol{G}\right)^{-1} \left(\bar{\xi}^T \boldsymbol{G} + \boldsymbol{G}^T \bar{\xi}\right) + \boldsymbol{G} \left(\boldsymbol{G}^T \boldsymbol{G}\right)^{-1} \left(\bar{\xi}^T \bar{\xi}\right) \right]^i \tag{46}$$

The same reasoning is used to deduce Eq. 12.

### I.3. Metrics related by scaling and constants

If $g$ is a metric and $g_{\text{scaled}} = Cg + D$, then from Eq. 40 and Eq. 42 we see that the geodesics induced by $g_{\text{scaled}}$ are the same as those induced by $g$. The geodesic sharpness induced by $g_{\text{scaled}}$ is

$$S_{\max}^\rho(w) = \mathbb{E}_{\mathbb{S}\sim\mathbb{D}} \left[ \max_{||\bar{\xi}||_{\bar{\gamma}_{\text{scaled}}} \leq \rho} L_S(\bar{\gamma}_{\bar{\xi};\text{scaled}}(1)) - L_S(\bar{\gamma}_{\bar{\xi};\text{scaled}}(0)) \right] =$$

$$= \mathbb{E}_{\mathbb{S}\sim\mathbb{D}} \left[ \max_{C||\bar{\xi}||_{\bar{\gamma}} + D \leq \rho} L_S(\bar{\gamma}_{\bar{\xi}}(1)) - L_S(\bar{\gamma}_{\bar{\xi}}(0)) \right],$$

$$= \mathbb{E}_{\mathbb{S}\sim\mathbb{D}} \left[ \max_{||\bar{\xi}||_{\bar{\gamma}} \leq \rho'} L_S(\bar{\gamma}_{\bar{\xi}}(1)) - L_S(\bar{\gamma}_{\bar{\xi}}(0)) \right],$$

So they are the same up to some re-definition of the hyperparameter $\rho$.

