# OpenReview forum: "Hide & Seek: Transformer Symmetries Obscure Sharpness & Riemannian Geometry Finds It"
_ICML.cc/2025/Conference — ICML 2025 spotlightposter_

### Official Review · Reviewer_EGqd · 2025-03-09

**Overall Recommendation:** 4

**Summary:**

The paper demonstrates that sharpness metrics on transformers are not a reliable proxy for generalization due to the symmetry properties of the attention mechanism. The author proposes using a Riemannian space, specifically a quotient manifold derived from the symmetry group. Within this space, they introduce a geometric sharpness metric and show that, for diagonal networks, an analytical solution exists. Experimental results on diagonal networks and Vision Transformers (ViTs) reveal a strong correlation between generalization and the sharpness metric. However, the correlation is less significant for language models.

**Claims And Evidence:**

The main claim is that classical sharpness is not well-suited for transformers due to specific symmetries in the parameter space. The second claim is that adapting the sharpness calculation to the quotient space, with respect to the symmetry group, improves its correlation with generalization. Both claims are supported theoretically and experimentally. The only reservation is that adaptive sharpness still performs reasonably well for vision models in the experimental results, although the proposed method is significantly better.

**Essential References Not Discussed:**

No

**Experimental Designs Or Analyses:**

I checked the validity of the three experiments. The protocols are straightforward and do not exhibit any issues.

**Methods And Evaluation Criteria:**

The evaluation follows protocols described in various papers. These methods are applied not only to the limited case of diagonal networks but also to pre-trained ViTs and language models. While deeper experimentation would be beneficial, the proposed method and evaluation criteria are well-founded.

**Other Comments Or Suggestions:**

- The Table 1 should be in the paper

**Other Strengths And Weaknesses:**

**Strengths:**
- The paper is very well written despite the complexity of the domain.
- The proposed approach is elegant and general enough to open important perspectives for generalization estimation using sharpness.

**Weaknesses:**
- The paper frequently refers to the appendix, making it more challenging to read.
- The experimental section is convincing but limited in terms of architectures.

**Questions For Authors:**

- What is the complexity and the computation cost of estimating geodesic sharpness? How does it compare to classical and adaptive sharpness? These questions are not sufficiently addressed in the paper and are not explored in the experimental section.

- How sensitive is the approach to different architectures? Only one architecture is considered for each experimental task, which could introduce bias in the results.

**Relation To Broader Scientific Literature:**

The paper is well-situated within the current literature, and the authors demonstrate a strong understanding of the state of the art in the field.

**Theoretical Claims:**

read the proofs but did not verify their correctness.

---

> ### Author Rebuttal · Authors · 2025-03-31
>
> We wish to thank the reviewer for their comprehensive review and for their helpful suggestions.
>
> **Other Comments Or Suggestions:**
> > The Table 1 should be in the paper
>
> Thank you for pointing this out. We will move it to the main body of the paper.
>
> ---
>
> **Questions**
>
> > Q1: What is the complexity and the computation cost of estimating geodesic sharpness? How does it compare to classical and adaptive sharpness?
>
> This is a good question!
> We will add the following to the paper:
>
> Geodesic sharpness does not significantly differ in time complexity from adaptive sharpness (or classical sharpness with a nearly identical time complexity).
> In our language model experiments, **adaptive sharpness takes 1.5s per step, while our $S_{\text{inv}}$ takes 3s, and our $S_{\text{mix}}$ around 2s per step**.
> For $S_\text{inv}$, the main additional overhead is inverting a $d_{\text{head}} \times d_{\text{head}}$ matrix and performing a Sylvester solve in SciPy on CPU, as there is no such solve in PyTorch.
>
> >Q2: How sensitive is the approach to different architectures? Only one architecture is considered for each experimental task, which could introduce bias in the results.
>
> This is indeed a limitation of the experimental setup we inherit from [1].
> To facilitate the comparison with [1], we decided to focus on the same architectures that were present there, but this could have a hidden bias.
> The reason for which this is done in [1] is to always compare models within the same loss surface (albeit at different points).
> Since the existence of correlation with generalization exists for all tasks we study, we suspect that this is not architecture-dependent, but are conducting further experiments with a broader set of ViT models to determine whether this indeed is the case.
>
> ---
>
> We hope we have addressed all of your questions and look forward to any further questions and insights that might come up during the discussion period.
>
> [1] Andriushchenko, M., Croce, F., Müller, M., Hein, M., and Flammarion, N. A modern look at the relationship between sharpness and generalization. 2023.

---

> > ### Comment · Reviewer_EGqd · 2025-04-02
> >
> > Thank you for the precise response—it confirms my positive assessment of the paper. However, I would appreciate a more in-depth comparison with the paper cited by the first Reviewer.

---

> > > ### Author Response · Authors · 2025-04-02
> > >
> > > Thank you for your response. Certainly --  we initially weren't aware of that paper, but will add a discussion about it in the related works section. To summarize, [1] introduces a quotient manifold construction for re-scaling symmetries and then use the Riemannian spectral norm as a measure of worst-case flatness; they validate their approach both on synthetic data (where they check the invariance of their measure) and on real-life data/models: MNIST and CIFAR-10; CNNs as models. The main differences from our approach are as follows:
> > > - a) Our approach is more general and can accommodate both the $GL(h)$ symmetry of transformers, and the original re-scaling/scaling symmetry of convolutional/fully-connected networks, rendering it applicable to a wider range of modern architectures;
> > > - b) Our experimental setup is more challenging: we test on large-scale models (large transformers vs CNNs) and large-scale datasets (ImageNet vs CIFAR-10). Sharpness measures that account for re-scaling/scaling symmetries (e.g. adaptive sharpness) work quite well on CIFAR-10 and for CNNs and tends to break down on datasets like ImageNet and for transformers;
> > > - c) Conceptually, [1] defines worst-case sharpness on the usual norm-ball, appropriately generalized to the Riemannian setting, characterized by $|| \xi|| \leq \rho$ . We propose instead that the ball should be the one traced out by geodesics, to better respect the underlying geometry.
> > > In our ablations in appendix G (Figure (7) and Figure (8)) ignoring the geodesic component of our approach corresponds to the middle plots, which have notably lower Kendall-tau correlation values than those obtained by using a geodesic-based sharpness measure.
> > > - d) Performance-wise, we believe our approach is more efficient because it does not use the Hessian, and need only to use considerably cheaper gradients. [2] mentions that even in a fully optimized setting, Hessian vector products calculations are at least 2 to 4 times as expensive as gradient calculations, and require between two and three times as much memory.
> > >
> > > [1] Rangamani, A., Nguyen, N. H., Kumar, A., Phan, D., Chin, S. P., & Tran, T. D. (2021, June). A scale invariant measure of flatness for deep network minima. In ICASSP 2021-2021 IEEE International Conference on Acoustics, Speech and Signal Processing (ICASSP) (pp. 1680-1684). IEEE.
> > >
> > > [2] How to compute Hessian-vector products? https://iclr-blogposts.github.io/2024/blog/bench-hvp/

---

### Official Review · Reviewer_hcv3 · 2025-03-12

**Overall Recommendation:** 4

**Summary:**

The paper proposes to define the sharpness of the loss curve of neural networks via Riemannian geometry in order to account for symmetries in network parameters. While some reparameterization-invariant sharpness measures exist, they do not account for all possible symmetries in parameters, in particular not for the symmetries of attention layers in transformers. The paper provides an instantiation of the proposed geodesic sharpness for this type of symmetry in transformer architectures. The paper finds that geodesic sharpness correlates stronger with generalization than the previously proposed adaptive sharpness measure.

**Claims And Evidence:**

The paper's derivations of geodesic sharpness are sound. The experiments show a superior correlation of geodesic sharpness with generalization, compared to adaptive sharpness. Although the reparameterization-invariance follows from the derivation of geodesic sharpness, it would have been nice to empirically verify this, as well.

**Essential References Not Discussed:**

The paper discusses essential references, to the best of my knowledge.

**Experimental Designs Or Analyses:**

The experimental design is sound. The result that sharpness is negatively correlated with generalization for vision transformers is curious. A potential explanation might be the requirement of locally constant labels suggested by the analysis in [1]: their analysis suggests that sharpness only correlates with generalization if labels in representation space can be assumed to be locally constant, i.e., small perturbations of the representation do not change the true label ($P(y|x)\approx P(y|x+\xi)$). If for the vision transformer small perturbations in the representation should lead to strong changes in the true label, then one would expect a negative correlation between sharpness and generalization. Of course, this argument holds for relative sharpness, and thus might not be true for geodesic sharpness.

[1] Petzka, Henning, et al. "Relative flatness and generalization." Advances in neural information processing systems 34 (2021): 18420-18432.

**Methods And Evaluation Criteria:**

The evaluation criteria are sound, geodesic sharpness is compared to adaptive sharpness in terms of its correlation with generalization for diagonal networks and transformers. The empirical evaluation could be a bit more comprehensive in terms of architectures, benchmark datasets and baseline generalization measures, but since the focus of the paper is on the theoretical contribution, I find the amount of experiments adequate.

**Other Comments Or Suggestions:**

I have no further comments.

######### After Rebuttal ############

I maintain my positive assessment and recommend acceptance.

**Other Strengths And Weaknesses:**

The paper tackles an important issue in research on the relationship between sharpness of the loss surface and generalization, namely that of symmetries in parameter space. The contributions are sound and original. While more empirical evaluation is required to show the practical significance of the proposed geodesic sharpness, its significance from a theoretical side is solid.

**Questions For Authors:**

Q1: Have you considered measuring average geodesic sharpness, as well?
Q2: Are the symmetries in [4] only instances of scaling and re-scaling?

[4] Petzka, Henning, Martin Trimmel, and Cristian Sminchisescu. "Notes on the symmetries of 2-layer relu-networks." Proceedings of the northern lights deep learning workshop. Vol. 1. 2020.

**Relation To Broader Scientific Literature:**

While Maksym Andriushchenko and his co-authors have shown that many sharpness measures do not correlate well with generalization (in particular the SAM-based ones), relative flatness [1] appears to work better, also with transformers, although no direct comparison has been made - to the best of my knowledge. That is, regularizing with relative flatness improves generalization also for transformers [2] and its behavior wrt. adversarial examples is similar-ish for CNNs and transformers [3]. Since computing it for the penultimate layer and the CE loss is very efficient [3], it would be interesting to discuss geodesic sharpness wrt. relative sharpness.

[2] Adilova, Linara, et al. "FAM: Relative Flatness Aware Minimization." Topological, Algebraic and Geometric Learning Workshops 2023. PMLR, 2023.
[3] Walter, Nils Philipp, et al. "The uncanny valley: Exploring adversarial robustness from a flatness perspective." arXiv preprint arXiv:2405.16918 (2024).

**Theoretical Claims:**

I have checked the theoretical claims and proofs. The proofs are presented clearly, up to my limited understanding of Riemannian geometry. Both claims and proofs are sound.

---

> ### Author Rebuttal · Authors · 2025-03-31
>
> We wish to thank the reviewer for their thoughtful review and their really interesting suggestions.
> We had not fully considered the possible connections with relative flatness, but find these to potentially be a really fruitful avenue of research.
>
> **Claims And Evidence:**
> >Although the reparameterization-invariance follows from the derivation of geodesic sharpness, it would have been nice to empirically verify this, as well.
>
> This is a good suggestion, and we will add this to any final versions in an appendix (similar to Figure 1 in Andriushchenko et al.).
>
> **Experimental Designs Or Analyses:**
> > The result that sharpness is negatively correlated with generalization for vision transformers is curious. [...]
>
> Thank you for this great observation! This could quite possibly help explain this curious phenomenon.
> One possible way to test for this (at least on synthetic datasets) would be to use a similar experimental setup to that used in [1], modified for classification with diagonal networks, where we can artificially control local label constancy through class separation.
> More broadly, and with an eye to possible further experiments, if the reviewer happens to be aware of any possibly useful approximations to the local constancy of labels, we would welcome any such suggestions.
> In principle, we need access to the data-generating process details to estimate the labels' local constancy, something we don't have for ImageNet.
>
> We also suspect that looking into the data distribution is the most promising future direction for understanding the sign of the correlation flip.
> For instance, in our synthetic regression task, we observe differing behaviours for the correlation between sharpness and generalization in the under or overparametrized regimes.
> The data itself can introduce additional symmetries in the overparametrized regime since in this regime $n<d$, where $n$ is the number of data points and $d$ is the dimension, and so the data matrix $X$, which is $n \times d$,  always has a non-trivial null space by the rank-nullity theorem.
> This implies that two predictors $\beta = u \odot v, \beta' = u' \odot v'$ should be equivalent if there is $z \in Null(X)$ s.t. $\beta=\beta'+z$.
> In the underparametrized regime, where $n>d$, this is no longer necessarily the case, and we expect the null-space to be trivial, thus making these symmetries disappear.
> We are unsure at this moment of what concrete impact these additional symmetries have but we intend to investigate them further.
> More broadly, data-dependent symmetries are already known in the literature (e.g. [2]), but remain, in our opinion, underexplored.
>
>
> **Relation To Broader Scientific Literature:**
> > That is, regularizing with relative flatness improves generalization also for transformers [2] and its behaviour wrt. adversarial examples is similar-ish for CNNs and transformers [3]. Since computing it for the penultimate layer and the CE loss is very efficient [3], it would be interesting to discuss geodesic sharpness wrt. relative sharpness.
>
> Thank you for the suggestion. We agree it would be interesting to add this to the paper, and we'll endeavour to do so in any final version.
>
> ---
> **Questions**
> > Q1: Have you considered measuring average geodesic sharpness, as well?
>
> This is a good question! We have considered it, especially since [3] reports that, at least for diagonal networks, average sharpness can have different correlation signs with generalization (positively correlated vs anti-correlated and vice-versa).
> We did not present results on it in our paper due to its mathematical complexity: we would need to find a computationally feasible algorithm to properly integrate over the high-dimensional geodesic ball.
> It was unclear to us whether this was entirely possible, although a Monte Carlo approach might yield results that are "good enough".
>
> **double-check**
>
> >Q2: Are the symmetries in [4] only instances of scaling and re-scaling?
>
> Exactly!
>
> ---
> We hope we have addressed all of your questions and look forward to any further questions and insights that might arise during the discussion.
>
> [1] Petzka, Henning, et al. "Relative flatness and generalization." Advances in neural information processing systems 34 (2021): 18420-18432.
>
> [2] Zhao, Bo, et al. "Symmetries, flat minima, and the conserved quantities of gradient flow." International Conference on Learning Representations 2023.
>
> [3] Andriushchenko, M., Croce, F., Müller, M., Hein, M., and Flammarion, N. A modern look at the relationship between sharpness and generalization. 2023.

---

> > ### Comment · Reviewer_hcv3 · 2025-04-02
> >
> > I thank the authors for their rebuttal.
> > - Regarding the problem of locally constant labels: I have not yet found a good solution for that myself. My main issue is that it seems that only a constant label in the representation is necessary, so even ensuring locally constant labels in the input distribution (e.g., via a synthetic dataset) does not guarantee locally constant labels in the representation. If using the penultimate layer, though, one might test for neural-collapse style clustering, since this implies - at least empirically - that labels are locally constant in representation space.
> > - I agree that investigating data-dependent (and maybe even data-distribution-dependent) symmetries is a very interesting direction for future research. I can only further encourage you to follow that path.
> >
> > I maintain very positive about this paper and keep my rating.

---

### Official Review · Reviewer_4oTA · 2025-03-13

**Overall Recommendation:** 3

**Summary:**

The paper introduces geodesic sharpness, a novel adaptive sharpness measure defined on a quotient manifold that factors out the rich symmetries in transformer parameter spaces (notably the high-dimensional GL(h) symmetry in attention). By leveraging Riemannian geometry, the authors redefine perturbation norms and paths (using geodesics) so that the measure is invariant to symmetry-induced redundancies. They show that when geodesic sharpness is approximated beyond first order, it recovers strong correlations with generalization, in contrast to traditional adaptive sharpness measures.

**Claims And Evidence:**

The main claims are that (a) existing sharpness measures fail in transformers due to unaddressed parameter symmetries, and (b) by reinterpreting sharpness on the quotient manifold, one can obtain a measure (geodesic sharpness) that correlates strongly with generalization. The evidence includes rigorous derivations for simple diagonal networks and empirical evaluations on vision transformers (fine-tuned CLIP) and language models (BERT fine-tuned on MNLI) where Kendall’s tau correlations are consistently stronger for geodesic sharpness. Overall, the claims are well supported, though the variability in the sign of the correlation across tasks suggests further investigation is warranted.

**Essential References Not Discussed:**

There are no missing references to my knowledge.

**Experimental Designs Or Analyses:**

The experimental setup is well thought out.

**Methods And Evaluation Criteria:**

The method reformulates the sharpness measure by defining the quotient manifold induced by network symmetries, lifting Euclidean objects to their Riemannian counterparts, and approximating geodesic paths to measure worst-case loss variation within a geodesic ball.
Evaluation is based on the correlation between the sharpness measure and generalization gap across controlled toy experiments and real-world transformer settings. The approach is conceptually sound, though practical geodesic approximations may require careful tuning.

**Other Comments Or Suggestions:**

A more detailed ablation study on the effect of metric choice (invariant vs. mixed) could help clarify when one might be preferred over the other.
Expanding the discussion on the conditions under which geodesic sharpness may flip its correlation sign would benefit practitioners.

**Other Strengths And Weaknesses:**

N/A

**Questions For Authors:**

N/A

**Relation To Broader Scientific Literature:**

The work extends the literature on generalization by connecting sharpness measures with the geometry of parameter space. By addressing higher-dimensional symmetries inherent in transformer architectures, it bridges a gap between geometric approaches in optimization and practical issues in modern deep learning.

**Theoretical Claims:**

While the derivations are largely convincing, the reliance on approximations and assumptions about the quotient manifold’s structure in complex networks are points that could benefit from further clarification.

---

> ### Author Rebuttal · Authors · 2025-03-31
>
> We thank the reviewer for their thorough review and their suggestions for improving the paper. We'll endeavour to include as much as possible in any final version.
>
> **Theoretical Claims**
>
> > While the derivations are largely convincing, the reliance on approximations and assumptions about the quotient manifold’s structure in complex networks are points that could benefit from further clarification.
>
> Thanks for pointing this out! We'll include a more detailed discussion on this in future versions.
>
> **Other Comments Or Suggestions**
>
> > A more detailed ablation study on the effect of metric choice (invariant vs. mixed) could help clarify when one might be preferred over the other.
>
> This is indeed a philosophically interesting question.
> From a theoretical perspective, we do not have any reason to prefer one metric over another as long as both of them correctly reflect symmetries.
>
> In practice, the metrics perform very similarly, with the mixed metric tending to perform slightly better and being faster to run. The mixed metric's numerics are also advantageous as we do not need to solve a Sylvester equation to project into the horizontal space.
>
> >Expanding the discussion on the conditions under which geodesic sharpness may flip its correlation sign would benefit practitioners.
>
> Thank you for this important question!
> This is something we're keen on investigating in future work, as it is one of the main factors limiting our approach's utility during training.
> We believe this will require significant research and taking into account more aspects, e.g. the data distribution (e.g. hypothesized by Reviewer hcv3), which are currently not considered in our framework that purely focuses on parameter space symmetries.
>
> ---
> We hope we have addressed all of your concerns and look forward to discussing any outstanding concerns in the discussion period.

---

### Official Review · Reviewer_jPWt · 2025-03-17

**Overall Recommendation:** 2

**Summary:**

This paper investigates the connection between sharpness and generalization for models with self attention layers by properly accounting for symmetries present in the models. The authors consider the quotient manifold of parameters and measure sharpness within a geodesic ball on the quotient manifold. The paper claims to introduce the application of Riemannian geometry to deep network parameter symmetry, the notion of geodesic sharpness, and solve for geodesic sharpness in diagonal networks and measure it empirically in transformers.

**Claims And Evidence:**

The theoretical and empirical results for diagonal networks seem contradictory. The analytical derivation suggests that when the estimated predictor is close to the optimal predictor, the sharpness should be small (Since $S \propto \|\| \beta_0 - \beta_* \|\|_2$). The empirical results show instead that larger sharpness is correlated with smaller test error, which means the estimated predictor is not close to the optimal predictor. Something does not seem right here.

The empirical results in image transformers and language models seem contradictory. Sharpness is correlated with better performance for image transformers and anti-correlated for LMs. Is this actually a meaningful correlate of generalization?

**Essential References Not Discussed:**

The authors claim that they are the first to apply Riemannian quotient manifolds to study deep network parameter symmetry. This was already done in a prior paper [1] that proposes a quotient manifold (along with a metric) for rescaling symmetries in MLPs/CNNs. The authors do not cite or discuss the relationship of their paper to this one.

[1] Rangamani, A., Nguyen, N. H., Kumar, A., Phan, D., Chin, S. P., & Tran, T. D. (2021, June). A scale invariant measure of flatness for deep network minima. In ICASSP 2021-2021 IEEE International Conference on Acoustics, Speech and Signal Processing (ICASSP) (pp. 1680-1684). IEEE.

**Experimental Designs Or Analyses:**

No apparent issues.

**Methods And Evaluation Criteria:**

The authors discuss LoRA adapters but do not conduct experiments for this scenario. The geodesic sharpness measure proposed by the authors could also be adapted to Residual networks that contain GL(h) symmetries within a residual block. Including these experiments would make the paper stronger.

**Other Comments Or Suggestions:**

None

**Other Strengths And Weaknesses:**

While the paper is easy to follow, it is unclear what the implications of its findings are. Are we able to leverage geodesic sharpness during optimization to find better minima? Can we prove tighter generalization bounds?

What is the time complexity of finding the geodesic sharpness? Why do they choose the algorithm in appendix C instead of a Riemannian Hessian based measure? How do the two compare?

**Questions For Authors:**

1. Can you reconcile your empirical and theoretical findings for the case of diagonal networks? I am confused how larger sharpness means better test loss

2. Please consider citing [1] since it already introduces a Riemannian quotient manifold of NN parameters albeit for rescaling symmetries.

3. What is the relationship between sharpness and generalization you can report? Your results seem to contradict the conventional wisdom that flat minima generalize better.


[1] Rangamani, A., Nguyen, N. H., Kumar, A., Phan, D., Chin, S. P., & Tran, T. D. (2021, June). A scale invariant measure of flatness for deep network minima. In ICASSP 2021-2021 IEEE International Conference on Acoustics, Speech and Signal Processing (ICASSP) (pp. 1680-1684). IEEE.

**Relation To Broader Scientific Literature:**

This paper proposes a quotient manifold and metric for GL(h) symmetries in deep network models that go beyond rescaling symmetries. This claim of theirs is accurate.

**Theoretical Claims:**

Skimmed through the Diagonal networks sharpness derivation, seems correct to me though I did not check details.

---

> ### Author Rebuttal · Authors · 2025-03-31
>
> Thank you for the insightful review and helpful suggestions for improvement.
> We will make sure to mention the reference [1] that you brought to our attention, and we will contrast it to our paper which goes beyond the re-scaling symmetry.
> Please find our answers to your remaining concerns below.
>
> **Q1**
>
> > Can you reconcile your empirical and theoretical findings [...]
>
> Thank you for pointing this out! There is indeed some context missing for it to fully make sense, but these findings are not contradictory.
>
> **TL;DR: The theoretical derivation (Equation (13)) assumes the underparameterized regime (less parameters than data), while the empirical results (Figure (3)) are for the practically more relevant overparameterized regime (more parameters than data), where deriving a closed-form is intractable.**
>
> Here are the details:
>
> - **Theoretical assumptions break down.**
>   The theoretical expression (Equation (13)) assumes that $X^\top X = I_{d\times d}$ with data matrix $X \in \mathbb{R}^{n \times d}$ (number of data $n$ and parameters $d$).
>   For this to be feasible, $n\geq d$, i.e. we are in the underparameterized regime.
>   This assumption allows us to derive closed-form expressions for the unique optimal predictor and its sharpness (see, e.g. [2]).
>   Unfortunately, analyzing the practically more relevant overparameterized regime exactly is intractable.
>
> - **We can reconcile empirical results with the theoretical prediction.**
>   If we re-run the experiment from Figure (3) (overparameterized regime) in the **underparameterized regime**, we indeed obtain the positive correlation between sharpness and generalization, **as predicted by the theory in Equation (13)**.
>   We believe it is practically less interesting because large models typically operate in the overparameterized regime.
>
> - **Our presented empirical findings are consistent with other works.**
>   The findings we report for overparameterized diagonal networks in the original submission agree with findings from other works.
>   The anti-correlation of worst-case sharpness with generalization was also found in [3].
> ---
>
> **Q2**
> > What is the relationship between sharpness and generalization you can report?
>
> Thank you for this important question!
> What we can report is that contrary to [3], once we account for symmetry, **there is a relationship between sharpness and generalization, and the Kendall-tau significantly differs from zero**. This is what we show in our experiments and we will make sure to make this more explicit in the text.
> We believe that answering how the different correlation signs come into being is beyond the scope of our paper.
> The full story is more complicated and rightfully deserves future investigation:
>
> Based on very recent insights from [4] which reports that sharpness minimization differs significantly between vision and language tasks, we hypothesize that the data distribution could play a significant role.
> Also, Reviewer hcv3 pointed out one interesting avenue to understanding this change in correlation sign via stability of the labels that we hope to follow up on in the very near future.
>
> **Other strengths and weaknesses**
>
> >While the paper is easy to follow, it is unclear what the implications of its findings are [...]?
>
> Our findings provide a foundation for future practical applications. Understanding the sign of the correlation between sharpness and generalization is the limiting factor, but once that is done, we expect geodesic sharpness to be useful for regularizing training.
>
> > What is the time complexity [...]?
>
> This is a good question!
> - **Computational cost:**
>    Geodesic sharpness does not significantly differ in time complexity from adaptive sharpness.
>   **In our language model experiments, adaptive sharpness takes 1.5s per step, while our $S_{\text{inv}}$ takes 3s, and our $S_{\text{mix}}$ takes around 2s**.
>
> - **Why we do not consider the Riemannian Hessian:**
> Mainly for reasons of computational efficiency. We deal with models with upwards of $100 M$ parameters, and even accessing quantities such as the Hessian trace through multiple Hessian-vector products is computationally expensive (this was already the case in [3]).
>
> ---
>
> **References:**
>
> [1] Rangamani, A., Nguyen, N. H., Kumar, A., Phan, D., Chin, S. P., & Tran, T. D. (2021, June). A scale invariant measure of flatness for deep network minima. In ICASSP 2021-2021 IEEE International Conference on Acoustics, Speech and Signal Processing (ICASSP) (pp. 1680-1684). IEEE.
>
> [2] Roger Grosse (2022). "A Toy Model: Linear Regression". University of Toronto, Topics in Machine Learning:  Neural Net Training Dynamics.
>
> [3] Andriushchenko, M., Croce, F., Müller, M., Hein, M., and Flammarion, N. A modern look at the relationship between sharpness and generalization. 2023.
>
> [4] Sidak Pal Singh, Hossein Mobahi, Atish Agarwala, and Yann Dauphin. Avoiding spurious sharpness minimization broadens applicability of SAM. arXiv preprint arXiv:2502.02407, 2025.

---

### Decision · Program_Chairs · 2025-05-01

**Decision:**

Accept (spotlight poster)

**Comment:**

This paper argues that understanding the link between flat minima and generalization is hindered by continuous parameter symmetries inherent in common architectures. To address this ambiguity, the authors propose using Riemannian geometry on the quotient manifold induced by these symmetries. Based on this framework, they introduce geodesic sharpness, a new adaptive sharpness measure where perturbations follow symmetry-respecting geodesic paths using the quotient manifold's metric. This method theoretically accounts for symmetry-induced curvature ignored by prior adaptive measures. Empirically, particularly when applied to the complex GL(h) symmetry within attention mechanisms, geodesic sharpness exhibits significantly stronger correlations with generalization performance on large transformers and language models than previously reported measures.

Reviewers generally acknowledged the importance of addressing parameter symmetries and found the proposed geodesic sharpness approach using Riemannian geometry to be theoretically sound, principled, and elegant. Several reviewers highlighted the potential significance of the work and noted that the empirical results, particularly on vision transformers, showed a stronger correlation between geodesic sharpness and generalization compared to adaptive sharpness. Two reviewers explicitly recommended acceptance, with another leaning towards acceptance.

Despite the positive reception, reviewers raised several concerns. A key point of discussion involved contradictory empirical results: the correlation between sharpness and generalization varied in sign between diagonal networks (theory vs. empirical overparameterized regime) and across tasks (vision vs. language models). Reviewer jPWt initially questioned the novelty claim regarding Riemannian quotient manifolds, pointing out a prior work (Rangamani et al., 2021) focused on rescaling symmetries, which the authors agreed to cite and differentiate from their more general approach. Other questions focused on the computational cost (which authors clarified is comparable to adaptive sharpness), the need for more extensive experiments across different architectures and datasets, the lack of empirical verification for reparameterization-invariance, and the need for further investigation into why the correlation sign flips between tasks. The authors provided detailed rebuttals, clarifying theoretical assumptions versus practical regimes, discussing potential reasons for correlation sign flips (hypothesizing data distribution roles and label stability), comparing their method's generality and efficiency to the cited prior work.

Overall, the assessment leans towards acceptance and in agreement with the reviewers, I believe addressing parameter symmetries is an important problem, and the proposed geodesic sharpness provides an interesting and principled approach. Therefore, I recommend acceptance. Please incorporate the suggested additions and clarifications into the final version.